# Polarized microtubule remodeling transforms the morphology of reactive microglia and drives cytokine release

Max Adrian [1,2], Martin Weber[1], Ming-Chi Tsai [1], Caspar Glock[3], Olga I. Kahn[1], Lilian Phu[4], Tommy K. Cheung[4], William J. Meilandt[1], Christopher M. Rose [4] & Casper C. Hoogenraad [1] ✉

Microglial reactivity is a pathological hallmark in many neurodegenerative diseases. During stimulation, microglia undergo complex morphological changes, including loss of their characteristic ramified morphology, which is routinely used to detect and quantify inflammation in the brain. However, the underlying molecular mechanisms and the relation between microglial morphology and their pathophysiological function are unknown. Here, proteomic profiling of lipopolysaccharide (LPS)-reactive microglia identifies microtubule remodeling pathways as an early factor that drives the morphological change and subsequently controls cytokine responses. We find that LPS-reactive microglia reorganize their microtubules to form a stable and centrosomally-anchored array to facilitate efficient cytokine trafficking and release. We identify cyclin-dependent kinase 1 (Cdk-1) as a critical upstream regulator of microtubule remodeling and morphological change in-vitro and in-situ. Cdk-1 inhibition also rescues tau and amyloid fibril-induced morphology changes. These results demonstrate a critical role for microtubule dynamics and reorganization in microglial reactivity and modulating cytokine-mediated inflammatory responses.

Microglia are tissue-resident macrophages of the brain and play important roles in maintaining tissue homeostasis[1]. Depending on their environment, microglia have various roles ranging from synaptic pruning and neuronal circuit remodeling in normal brain development to pathological effects in autism spectrum disorders, schizophrenia, and various neurodegenerative diseases[2,3]. Recently, several studies have begun to dissect the different activity states of microglia that could explain their variety of functions in brain disease[4,5]. Reactive microglia cluster at pathological sites where they execute protective functions involved in the removal of dead cells, protein aggregates and other cellular debris. On the other hand, the sustained or chronic overstimulation of microglia with excess production of inflammatory mediators can lead to detrimental neuroinflammation[6]. Histologically,

the most striking difference between homeostatic and reactive microglia is a distinct shift in cell morphology that has been used to measure microglial reactivity in patients and in vivo models for decades[7]. Importantly, microglial morphology is specific to the type of stimulus encountered and on their location within the brain[8,9]. However, the cell biological link between morphological and functional changes in microglia has not been studied comprehensively.

Within their tiled territories, homeostatic microglia have dynamic ramified protrusions that contact all other cell types in their vicinity and such extensive connections make resting microglia excellent sentinels surveilling the tissue for danger signals[10,11]. Microglial stimulation leads to a dramatic loss of cell protrusions and microglia become ameboid and motile[12]. Changes in cell morphology are

---

[1]Department of Neuroscience, Genentech, Inc., South San Francisco, CA 94080, USA. [2]Department of Pathology, Genentech, Inc., South San Francisco, CA 94080, USA. [3]Department of OMNI Bioinformatics, Genentech Inc., South San Francisco, CA 94080, USA. [4]Department of Microchemistry, Proteomics and Lipidomics, South San Francisco, CA 94080, USA. ✉e-mail: hoogenraad.casper@gene.com

consequences of rearrangements of the underlying actin and microtubule cytoskeletal organization. Until now, our understanding of the cytoskeletal dynamics in homeostatic and reactive microglia is very limited.

Microtubules are involved in cellular processes including motility, intracellular transport, and maintenance of cell shape in many cells in the CNS, including neurons and oligodendrocytes[13,14]. Microtubules have been shown to be essential for the branched morphology of microglia in vivo[10]. However very little is known about the role of microtubule dynamics during microglial reactivity[15]. Reactive microglia have microtubule organizations that are less dense than resting microglia in vitro. Their microtubules extend throughout the cytoplasm and get stabilized via posttranslational acetylation at the microtubule organizing center (MTOC)[16,17]. Most recently, Golgi outposts at branching points of microglial ramifications have been demonstrated to play a role in nucleating microtubules in homeostatic microglia[18]. Still, it remains unknown which factors and pathways are involved in microtubule organization during the process of microglial reactivity and how these cytoskeletal rearrangements relate to their function.

Here, we study the initial morphological change of reactive microglia, the underlying cytoskeletal rearrangements, and the link to inflammatory responses. We find that microglial cell morphology is dependent on dynamic microtubules that are highly regulated during stimulation. Microglia reorganize their microtubules into a radial array by activating their centrosomes and suppressing Golgi-mediated microtubule polymerization. Moreover, microtubule polymerization is favored in reactive microglia through downregulation of Stathmin 1 (Stmn1) and microtubule stabilization is mediated by microtubule-associated protein 4 (Map4). All three pathways are driven by the activation of cyclin-dependent kinase 1 (Cdk1), in absence of cell division. Cdk1 activity is required for the ameboid cell morphology of reactive microglia in acute brain slices and efficient trafficking and release of cytokines. Overall, this study highlights that microtubule remodeling is required to transform homeostatic microglia into a reactive ameboid state and drives cytokine-mediated inflammatory responses.

## Results

### Microglial morphology changes in late and early reactive states

Neuroinflammation has often been quantified by staining brain slices for microglial marker proteins, e.g., Iba1, and measuring staining intensity or positive areas over an empirically set threshold[19,20]. Recently more sophisticated machine-learning approaches have revealed nuanced regional differences in microglial morphology[21]. Alternatively, other studies have developed methods to segment individual microglial cells imaged with high-resolution microscopy to describe their morphology, especially their ramified protrusions, in ultrastructural detail[22]. We established a robust cell segmentation workflow based on widely used 3,3′-diaminobenzidine (DAB) staining of Iba1 protein in mouse brains (Fig. 1a–e, Supplementary Fig. 1a) allowing us to assess the overall changes in cell morphology for individually segmented microglia in situ. We resolved individual cell morphologies by first detecting individual cell bodies (Fig. 1b) and subsequent intensity-based water-shedding to separate protrusions of neighboring cells (Fig. 1c) (see "Methods" for details). Importantly, this routine excludes branches of cells that were not fully visible in the section analyzed, significantly reducing overestimation of the cell area seen in threshold-based approaches (Fig. 1d). A large range of cell morphologies, from ramified to ameboid, was detected and morphometric parameters were extracted for each cell (Fig. 1e). This analysis routine could in principle be adapted to other microglial stains, e.g., Tmem119-DAB yielding similar results (Supplementary Fig. 1b). As a particularly useful metric for cell reactivity in situ, we established the ramification index shown in Fig. 1e to differentiate highly ramified (close to 1) from less ramified cells (0.8 and lower) along with cell area and perimeter.

To test if our segmentation approach can detect microglial reactivity in neuro-inflammatory models, we applied it to images of microglia from well-characterized transgenic mouse models known to induce microglial reactivity. Microglia of transgenic mice expressing Tau[P301S][23,24] showed decreased ramification in the hippocampus as early as 6 months and more pronounced at 12 months of age (Fig. 1f, g, Supplementary Fig. 1c). These results demonstrate that even small differences in ramification can be reliably detected by averaging the results of thousands of cells per animal. Similarly, we could detect loss of ramification in microglia in an established model of beta amyloidosis dependent on expression of the microglial receptor Trem2[20]. PS2APP expression induced robust loss of microglial ramification at 6 months of age both in the hippocampus and cortex of Trem2[WT] mice. In Trem2[KO] mice microglia remained ramified demonstrating their deficit in reactivity to amyloid plaques (Fig. 1h, i, Supplementary Fig. 1c). This validation of our analysis pipeline gave us confidence about our sensitivity and showcases the range of morphologies observed under healthy and pathological conditions. Thus, microglial morphology assessed through averaged single-cell segmentation allows measuring their reactivity in Iba1-stained brain sections.

These genetic models of neurodegenerative diseases show that microglial reactivity leads to morphological changes at advanced pathological timepoints. We next determined whether cell morphology changes can be detected at very early timepoints in an acute systemic inflammation, modeled by dosing wild-type mice intraperitoneally (i.p.) with LPS[25]. Loss of ramification could be observed as early as 6 h post injection and became more apparent over two days (Fig. 1j, k, Supplementary Fig. 1c). This systemic inflammatory model induced a spike in the production of inflammatory cytokines in the blood plasma and in the brain tissue (Supplementary Fig. 1d), and caused an increase in the phagocytosis marker CD68 at later timepoints (Supplementary Fig. 1e). Together, these results show that morphological changes typically studied at late-stage pathological stages of neurodegeneration share morphological similarity with a model of short systemic inflammation leading to loss of ramification and reduction of cell size in situ.

### Microglial morphology is dependent on dynamic microtubules

To better understand the functional significance of microglia morphology changes, we sought to mimic them in an in vitro model that would be amenable to high-resolution imaging techniques and molecular tools. To do so, we established primary cultures of murine microglia that were >97% pure and maintain a microglial gene expression phenotype (Supplementary Fig. 2a, b). While these cultured primary cells do not fully mimic homeostatic microglial cells in the tissue, they retain their ability to be further polarized into inflammatory and anti-inflammatory states as assessed by qPCR and cytokine profiling (Supplementary Fig. 2c, d). Cultured microglia survived well in presence of CSF-1 but barely proliferated as tested by immunostaining for Ki67 and phosphorylated Histone H3. Remarkably, less than 1% of cells were positive for phosphorylated Histone H3, a marker for cells in prophase. Consistent with earlier reports, stimulation with LPS further reduced the number of mitotic cells to near zero (Supplementary Fig. 2e)[26].

While numerous reports have shown that cultured microglia are much less ramified than in vivo[27,28], we noted that under our culture conditions resting microglia showed an elongated cell morphology and moderate branching. The lack of extensive ramification in vitro is likely due to the lack of tissue structure, as ramification can be induced in co-cultures with astrocytes and neurons in vitro. Hence, in primary cells, we expected to see a smaller reduction in relative ramification index after stimulation compared to in situ studies, but instead focused on differences in cell size. Stimulation of cultured microglia

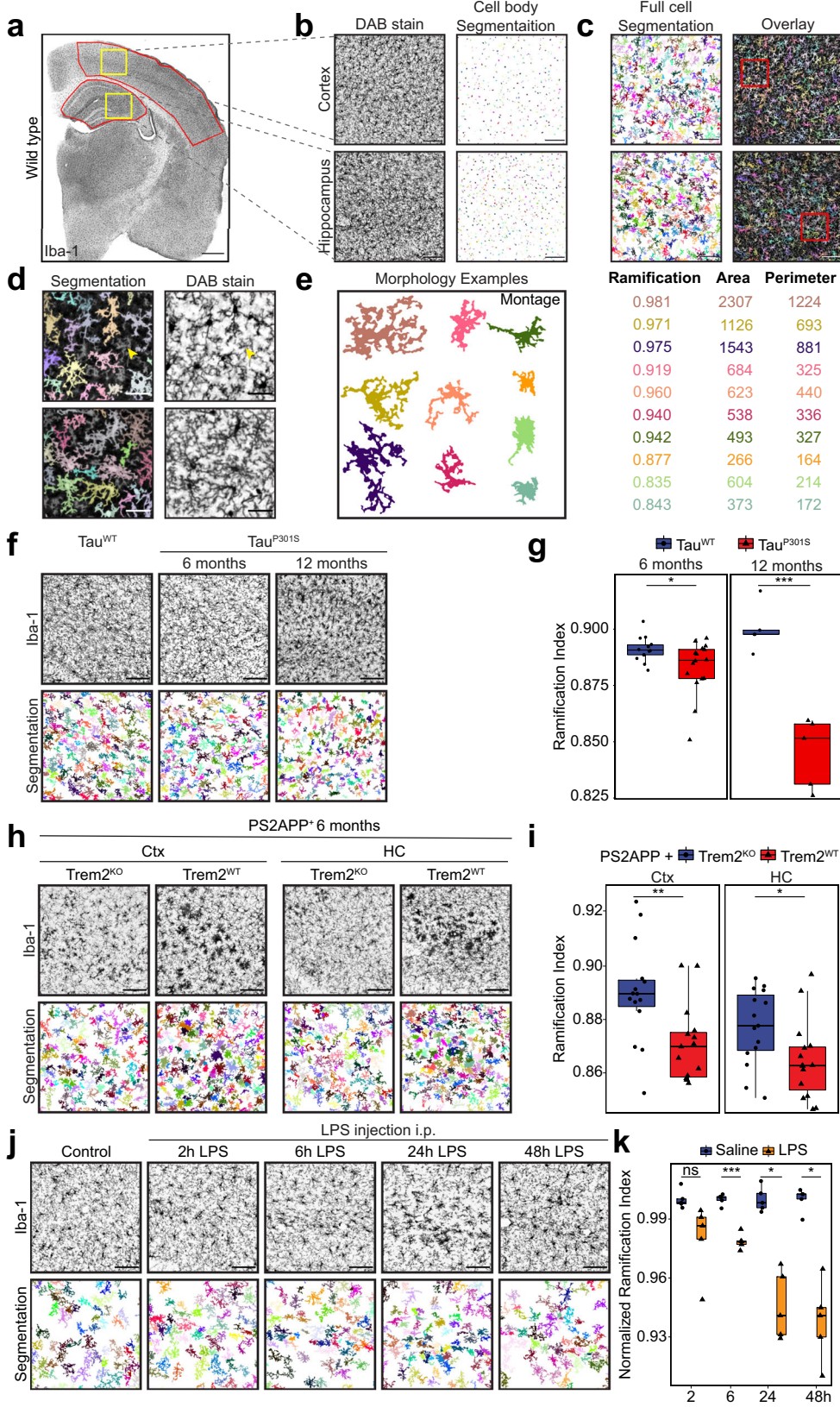

with LPS quickly affected their morphology and induced larger cell sizes and loss of ramification and increased cell perimeter within hours (Fig. 2a–d). We further analyzed microglial morphology after stimulation with cytokines or monomeric and fibrillated amyloid beta and

tau protein in vitro and noted an expected variance in the resulting morphology after 24 h. LPS stimulation elicited the strongest response mimicking the in situ results that we chose as a model for the following studies. Interestingly, incubations with recombinant beta amyloid and

**Fig. 1 | Reactive microglia show similar morphological changes in late-stage pathology and acute systemic stimulation models. a** Immunostaining of Iba1 in murine hemibrain of wild-type mouse. Cortex and hippocampal regions of interest that were subsequently quantified are highlighted in red boxes. **b–d** Single-cell segmentation workflow of microglia in mouse brain sections as shown in a. All steps are shown for regions highlighted in yellow boxes in a. Immunostaining and segmented cell bodies are shown in (**b**). Watershed-based segmentation individual ramified microglia cells and overlay with original staining is shown in c and d in higher magnification for the areas highlighted in red. Arrowheads in d show cell branches stemming from cells located outside the section stained that are purposefully omitted in the segmentation result. **e** Montage of representative cell morphologies illustrating ramified and ameboid cell morphologies along with a selection of their morphological descriptors. **f** Immunostaining for Iba1 and single-cell segmentation results in hippocampi of 6- and 12-month-old Tau$^{WT}$ and Tau$^{P301S}$ mice. **g** Quantification of ramification index of microglia measured in (**f**) per age and averaged per animal, $n = 14, 16, 5, 5$ animals per group. **h** Immunostaining for Iba1 and single-cell segmentation results in cortex (Ctx) and hippocampus (HC) of 6-month-old Trem2$^{WT}$ and Trem2$^{KO}$ animals crossed into PS2APP amyloidosis model. **i** Quantification of ramification index of microglia measured in h per brain region and averaged per animal, $n = 15$ animals per group. **j** Immunostaining for Iba1 and single-cell segmentation results of WT mice injected with LPS *i.p.* at timepoints indicated. **k** Quantification of ramification index of microglia measured in j averaged per animal, $n = 5$ animals per group. Scale bars are 500 μm in a, 100 μm in (**b, c, f, h, j**) and 30 μm in (**d**). Boxplots show all datapoints, median, 25th and 75th percentile, whiskers are 1.5*IQR. Statistical significance was calculated with unpaired, two-sided *t*-tests in (**g, i, k**). Significance intervals p: ****<1e−04 <***<0.001 <**<0.01 <*<0.05 <ns. Source data are provided as a Source Data file.

Tau$^{P301S}$ fibrils also affected cells size and/or ramification (Fig. 2a, b). Similarly, we observed that stimulation with Interleukin 4 (IL-4) and 13 or interferon gamma increased cell size and Tumor Necrosis Factor (TNF) alpha, interferon gamma or Chemokine ligand 2(CCL-2) reduced cell ramification. Treatments with IL-6 or IL-10 had no effect on microglial cell morphology (Supplementary Fig. 2f).

Importantly, we found that dynamic microtubules are required for this change in microglial morphology. Treating microglia with 1 μM taxol or nocodazole, drugs that hyper-stabilize and depolymerize microtubules, respectively, prevented the LPS-induced morphology changes (Fig. 2e, f). Moreover, the addition of the microtubule agents also reduced the amount of cytokines secreted from microglial cultures in a dose-dependent manner (Fig. 2g, Supplementary Fig. 2g). Overall, these experiments show that we have established an in vitro model to study early morphological changes and that reactive microglia are dependent on a dynamic microtubule cytoskeleton.

## LPS-reactive microglia extensively regulate inflammatory and cytoskeletal proteins

To better understand the processes that mediate microglial reactivity, we quantitatively analyzed the proteome and phospho-proteome of primary microglia treated with LPS in a time course study (Fig. 3a). Microglial LPS stimulation led to a very rapid regulation of 11, 30, 165, 485 and 908 proteins at 0.5, 1, 4, 8 and 24 h, respectively (Fig. 3b, Supplementary Data 1). Gene ontology (GO) analysis of the proteins upregulated after 8 h showed significant enrichment of cytoskeleton components as well as inflammatory and cytokine sensing and producing processes (Fig. 3c, d, Supplementary Data 2). Upregulation of inflammatory proteins as IL-1a, Ptgs2 and TNFa was very fast and pronounced. In contrast, proteins associated with microtubules only showed very moderate changes in protein levels (Fig. 3e, f). Microtubule dynamics are however not only controlled by the abundance of microtubule-associated proteins but also strongly affected by their posttranslational regulation, especially phosphorylation[29]. The number of differently regulated phosphorylated peptides was largest immediately after LPS treatment (1315 peptides at 30 min), but remained high for at least 4 h (1154 and 1169 peptides at 1 and 4 h, respectively) (Fig. 3g and Supplementary Data 1). The regulated proteins were enriched for cytoskeletal and microtubule components and involved in processes of organelle organization and protein transport amongst others (Fig. 3h, i and Supplementary Data 2). Indeed, we detected many phosphorylated peptides associated with GO terms "microtubule" and "microtubule organizing center", 30 min after LPS treatment (Fig. 3j), indicating that remodeling of the microtubule network may be an early and critical factor in microglial reactivity.

Importantly, we found robust phospho-regulation of two well-studied microtubule-regulating proteins: Map4 and Stmn1 (Fig. 3k). While Map4 protein levels were not changed in LPS-reactive cells, we detected strong upregulation of phosphorylation on several sites. Map4 is a microtubule stabilizer, that suppresses catastrophe events if bound to microtubules. Map4 binding to microtubules, in turn, is regulated through inhibitory phosphorylation in the Map4 microtubule-binding regions[30–32]. Stmn1 is another regulator of microtubule dynamics, that promotes microtubule instability and sequesters free tubulin dimers, preventing them from polymerizing[33]. Phosphorylation of Stmn1 S25 and S38, has been shown to target the protein for degradation by the proteasome[34,35]. Indeed, phosphorylation at these sites was detected immediately after LPS stimulation and consequently protein levels reduced over the following hours. Together these findings demonstrate that the microtubule cytoskeleton is strongly regulated during initial microglial reactivity. Moreover, identifying microtubule factors together with intracellular cytokine trafficking and release pathways further supported our initial observation that these two processes are strongly linked.

## LPS-reactive microglia polarize microtubules in a plus-end-out radial array

To determine the effect of microglial reactivity on microtubule dynamics, we first visualized microtubules by immunostaining for alpha-tubulin in unstimulated and LPS-stimulated microglia. Unstimulated microglia showed a dense microglial network throughout the cell with many microtubules bending and buckling in the cell periphery. In contrast, reactive microglia showed a strong radial polarization with microtubules extending straight from the centrosome into the cell periphery (Fig. 4a). As the dense network in unstimulated cells is hard to resolve even with super-resolution microscopy, we stained for EB1 and Camsap2, markers for growing plus tips and free stable minus-ends, respectively. We detected polymerizing plus-ends in both unstimulated and reactive microglia, however, Camsap2 signal was strongly reduced in reactive microglia (Fig. 4b). This indicates a reduction of free microtubule minus-ends that are often associated with non-centrosomal nucleation sites, e.g., the Golgi apparatus[36,37]. To quantify this shift in microtubule phenotypes, we classified cells into disorganized (as shown in Fig. 4a control) and radially polarized microtubule arrays (as shown in Fig. 4a LPS). Cells with radially polarized microtubules became more prevalent over 16 h of LPS treatment and then stabilized at ~60 % of the population, while unpolarized cells became less abundant (Fig. 4c). These data show that the overall microtubule cytoskeleton rearranges within hours of microglial stimulation.

We next determined the microtubule orientation in microglia. The radial array centered on the centrosome suggests a plus-end-out orientation in LPS-reactive microglia but this is difficult to analyze in fixed cells. Instead, we established microtubule plus-tip tracking in primary microglia. This technique uses the expression of microtubule plus-tip binding peptides fused to a fluorescent protein (MT + TIP) to track the displacement of growing microtubules with live cell microscopy (Supplementary Movie 1). We found that primary microglial cultures infect readily with single-stranded adeno-associated virus with an engineered serotype (AAV-DJ) yielding robust expression of

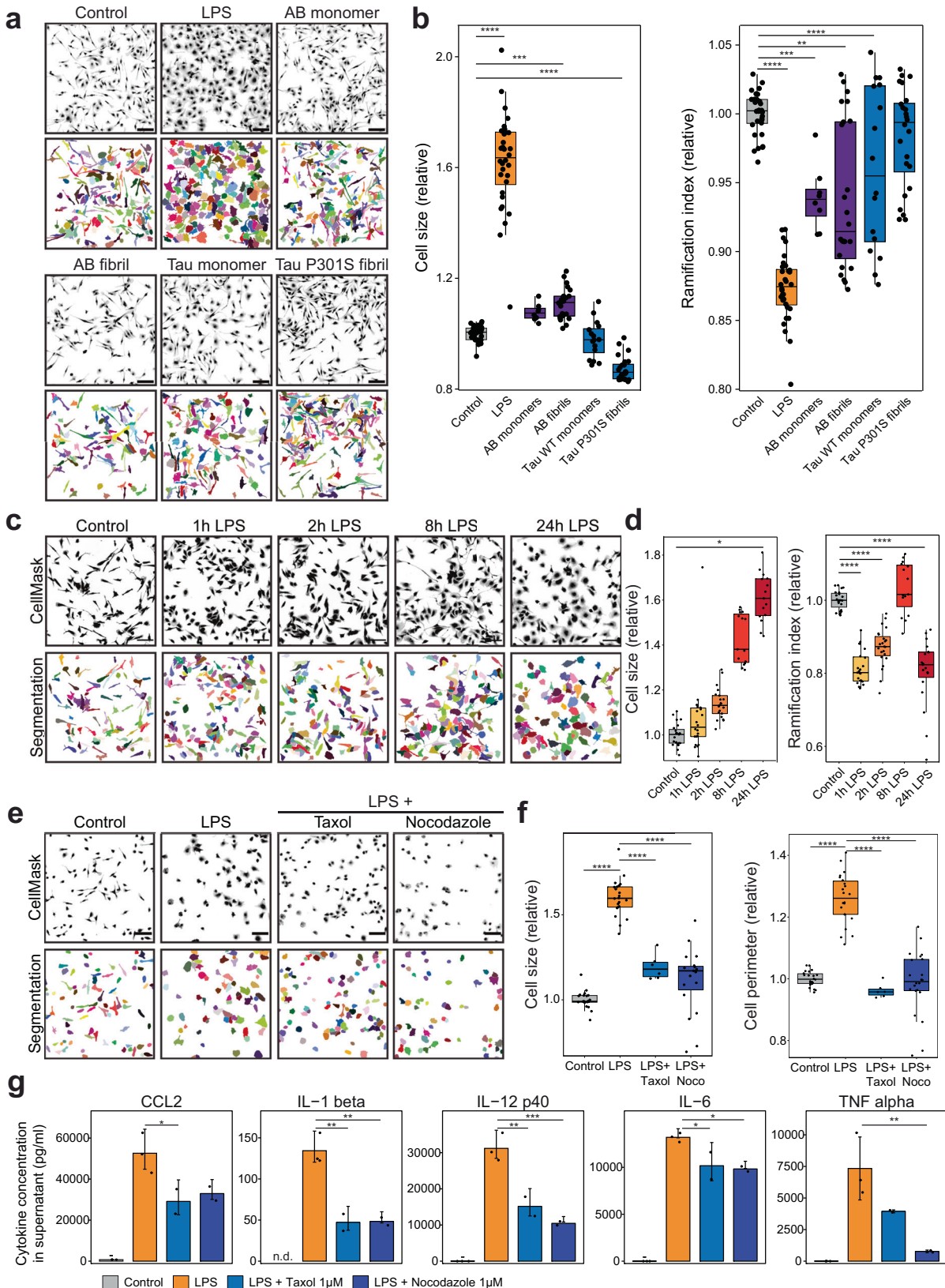

exogenous proteins within 1–2 days. Importantly, we did not detect upregulation of inflammatory marker gene expression in microglia infected with these viruses alone (Supplementary Fig. 3a). Tracking MT + TIP-eGFP in unstimulated and reactive microglia confirmed our hypothesis that microtubule dynamics change during stimulation (Fig. 4d, e). In particular, we detected fewer microtubules growing towards the cell center and fewer microtubule bending events at the plasma membrane in reactive microglia. In addition, reactive microglia had larger numbers of tracks measured, indicating an overall increase in microtubule polymerization. Together, these results show an increase in microtubule dynamics along with a more pronounced plus-tip-out orientation in LPS-reactive microglia.

**Fig. 2 | An in vitro model of microglial LPS-reactivity recapitulates morphological changes seen in situ and demonstrates dependence on dynamic microtubules. a** Representative images of primary microglia stained with CellMask dye and single-cell segmentation results after indicated stimulations for 24 h. **b** Quantification of cell size and ramification index of segmented results shown in (**a**) averaged per well relative to untreated cells, $n = 32, 32, 8, 24, 16, 24$ wells from 4 independent cultures. **c** Representative images of primary microglia stained with CellMask dye and single-cell segmentation results in a LPS stimulation time course. **d** Quantification of cell size and ramification index of segmented results shown in (**c**) averaged per well relative to untreated cells, $n = 23, 23, 23, 15, 22$ wells from 3 independent cultures. **e** Representative images of microglia stained with CellMask dye and single-cell segmentation results after treatment with LPS and microtubule

poisons Taxol and Nocodazole for 24 h. **f** Quantification of cell size and cell perimeter of segmented cells shown in (**e**) averaged per well relative to untreated cells, $n = 22, 22, 6, 22$ wells from 4 independent cultures. **g** Quantification of cytokines secreted into the supernatant of microglial cultures over 24 h after LPS and Taxol or Nocodazole treatment. Measurements out of detection range are indicated by n.d. $n = 3$ replicates. Scale bars are 100 μm. Bars indicate mean ± SD in (**g**), scatters show average values per well in (**b, d, f**) and replicates in (**g**). Boxplots show all datapoints, median, 25th and 75th percentile, whiskers are 1.5*IQR. Statistical significance was calculated with one-way ANOVA and Tukey HSD in (**b, d, f, g**). Significance intervals p: **** <1e−04 <*** <0.001 <** <0.01 <* <0.05 <ns. Source data are provided as a Source Data file.

## Microtubule nucleation is driven by the centrosome in LPS-reactive microglia

The increase in radial polarization of microtubules in LPS-stimulated microglia let us hypothesize that centrosomal activity may be upregulated leading to increased nucleation of microtubules in these cells. Indeed, gamma-tubulin and pericentrin, proteins involved in microtubule nucleation and centrosomal maturation, respectively, were more strongly enriched at centrosomes of microglia after LPS stimulation (Fig. 4f). Comparing centrosomal and cytoplasmic levels of these proteins, showed that gamma-tubulin was selectively lost from the cytoplasm, while pericentrin accumulated at the centromere (Fig. 4g). Overall protein levels of pericentrin did not change, while gamma-tubulin was overall downregulated 8 and 24 h after LPS stimulation (Supplementary Data 1 and Supplementary Fig. 3b). These results, together with the loss of Camsap2-positive minus-ends in the cell periphery (Fig. 4b), suggest that centrosomes mature and recruit more pericentriolar material (PCM) as nucleation of microtubules is concentrated at this site and lost in the periphery.

We investigated the nucleation sites in unstimulated and reactive microglia with nocodazole washout-studies. In this assay, microtubules were depolymerized by nocodazole treatment for 1 h. At this time no microtubules were detected in treated microglia. Washing out nocodazole for 1 min before fixation of the cells allows microtubules to regrow briefly at the sites that harbor the nucleating machinery (Fig. 4h). In unstimulated cells, many microtubules grew from extra-centrosomal sites and ~40% of these free microtubules emanated from GM-130 positive Golgi structures. In LPS-reactive cells, tubulin was more concentrated at the centrosome and significantly fewer free and Golgi-associated microtubules were found (Fig. 4h–j). Live imaging of MT + TIP in cells recovering from nocodazole treatment showed that MT + TIPs were found throughout resting microglia but confined to the centrosomal region in reactive cells (Supplementary Movie 2). Finally, we found that Akap9, a protein anchoring microtubule nucleation machinery to Golgi membranes was downregulated in LPS-reactive microglia (Supplementary Data 1, Supplementary Fig. 3c). Functionally, we assessed the significance of this microtubule rearrangement by measuring the release of cytokines. Reducing Golgi-associated microtubules through knockdown of *Akap9* led to increased release of cytokines in response to LPS treatment (Supplementary Fig. 3d, e). This result indicates that the polarization of microtubules is required for efficient cytokine release in reactive microglia. In summary, these data demonstrate that microtubule nucleation switches from nucleation at centrosomal and peripheral sites, including the Golgi apparatus, to nucleation exclusively driven by the centrosome in LPS-reactive microglia. This leads to the establishment of a radial microtubule array facilitating cytokine release.

## Stmn1 and Map4 pathways increase microtubule polymerization and stabilization

To get a better mechanistic understanding of the factors involved in rearranging the microtubule network in reactive microglia, we studied

tubulin posttranslational modifications and the Stmn1 and Map4 pathways identified in the phosphoproteomic data. First, immunoblots for tubulin and its posttranslational modifications showed only a slight increase in total tubulin levels, yet we see that the amount of acetylated tubulin doubles within 4 h of LPS stimulation. Levels of tyrosinated and detyrosinated tubulin did not change significantly (Fig. 5a, Supplementary Fig. 4a). We confirmed this increase in microtubule acetylation with immunofluorescence and found a strong increase in signal at the centrosomal microtubules in the perinuclear space (Fig. 5b, Supplementary Fig. 4b), suggesting that microtubule stabilization is an early event in reactive microglia cells. While the total levels of tubulin protein did not change significantly, we hypothesized that more tubulin was polymerized into microtubules in LPS-reactive microglia. Supporting this, we observed a decrease of Stmn1 protein levels, a negative regulator of microtubule growth in reactive microglia (Supplementary Data 1, Fig. 5c). Phosphorylation of Stmn1 at S25 and S28 were detected that are known to target Stmn1 for proteasomal degradation (Supplementary Data 1, Fig. 5d). To test if the ratio of soluble and polymerized tubulin changed as a result of Stmn1 downregulation, we performed microtubule pelleting assays. Indeed, LPS-reactive microglia showed a strong increase in the polymerized fraction. Overexpressing a non-phosphorylatable Stmn1$^{S25A\ S38A}$ mutant (Stmn1$^{2xSA}$) blocked this increase back to control levels (Fig. 5e, Supplementary Fig. 4c). In addition, we performed MT + TIP tracking in LPS-stimulated microglia overexpressing Stmn1$^{2xSA}$ and noted a reduction in MT + TIP track number, their speed and total displacement (Fig. 5f, g). Taken together, these data show that Stmn1 downregulation in LPS-reactive microglia lead to increased microtubule polymerization.

Our proteomic analysis highlighted strong phospho-regulation of the microtubule-stabilizing protein Map4 while its abundance did not change (Supplementary Data 1, Fig. 3f, k, Supplementary Fig. 4d). MAP binding to microtubules is known to be regulated by inhibitory phosphorylation in the microtubule domains[31]. Indeed, we found Map4 S914 and S1046, residues in microtubule-binding domains 3 and 4, to be less phosphorylated in reactive microglia, suggesting that Map4 may associate more with microtubules in this condition (Supplementary Data 1, Fig. 5h). In addition, we found strong upregulation of phosphorylation sites in the uncharacterized N-terminal domain of Map4 in reactive microglia that could potentially influence Map4 activity or microtubule binding through other mechanisms (Supplementary Data 1, Fig. 3k, Supplementary Fig. 4e). In immunofluorescence assays, Map4 was observed to associate with microtubules more strongly after LPS stimulation (Fig. 5i). We next investigated the role of Map4 in microglial reactivity by knocking it down with siRNA that resulted in loss of >75% *Map4* mRNA and loss of >85% of Map4 immunostaining (Supplementary Fig. 4f, g). In microtubule pelleting assays, *Map4* knockdown reduced the amount of polymerized tubulin, indicating that Map4 is required to promote microtubule assembly (Fig. 5j, Supplementary Fig. 4h). Knockdown of *Map4* in MT + TIP tracking assays revealed that Map4 slows down microtubule growth and reduces the number of tracks as well as their

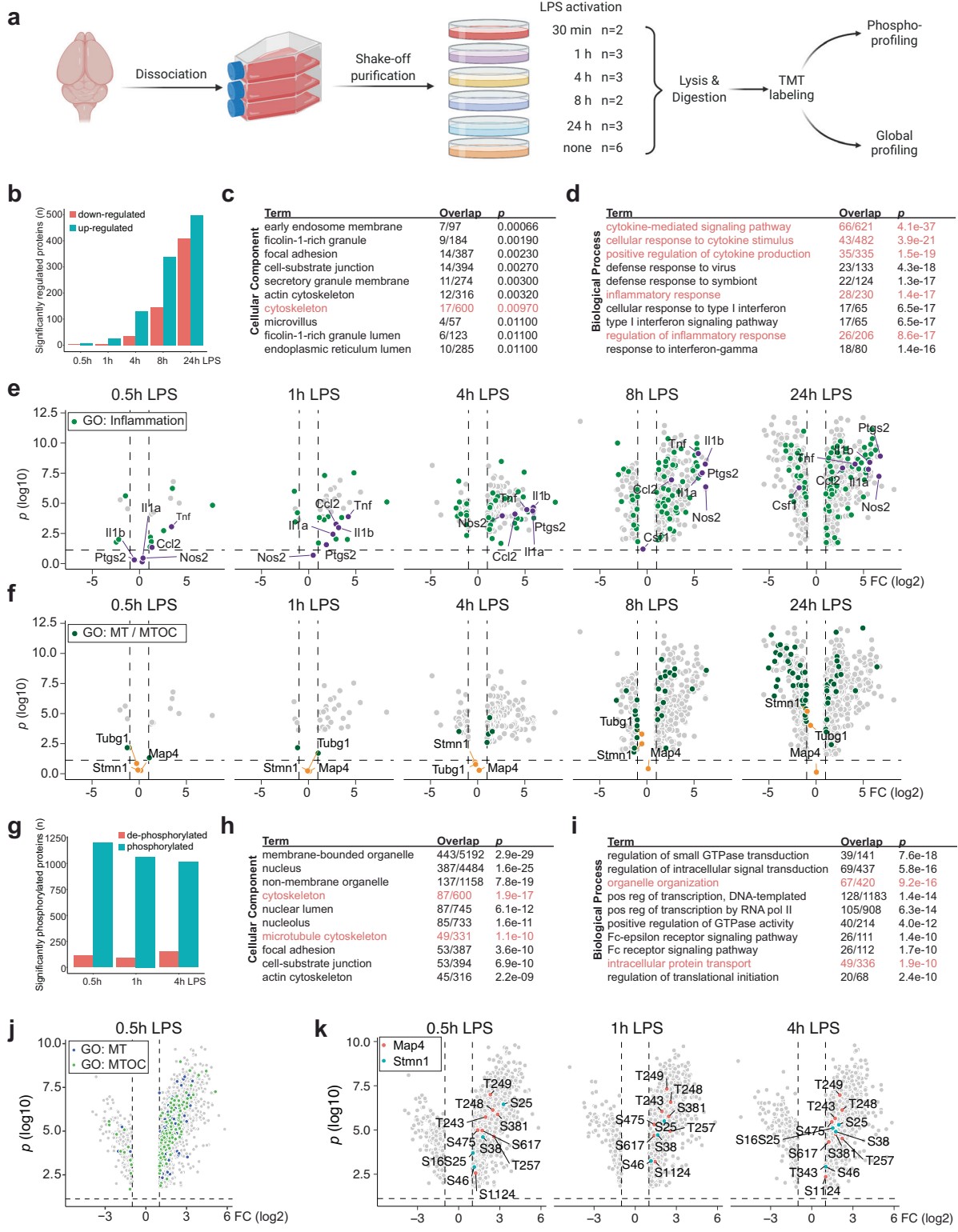

displacement while not affecting their direction, as expected (Fig. 5k, Supplementary Fig. 4i). In addition, knockdown of *Map4* also reduced the levels of acetylated and tyrosinated tubulin in reactive microglia (Fig. 5l). These data suggest that Map4 association with microtubules in reactive microglia contribute to the establishment and maintenance of stable microtubule arrays. The combined effect of Stmn1 down-regulation and Map4 activation contributes to rearranging and stabilizing the microtubule cytoskeleton within 1–4 h of stimulation with LPS.

## Cdk1 activation is the upstream signaling event causing microtubule rearrangement

To better understand the signaling pathway underlying the microtubule rearrangements in reactive microglia, we searched for upstream regulatory pathways by using the proteomic data of primary microglia treated with LPS (Fig. 3a). Phospho-proteomics revealed that LPS stimulation induced Cdk1 activation through loss of inhibitory phosphorylation on T14 and Y15 (Supplementary Data 1, Fig. 6a). Importantly, this stimulation did not induce mitosis or cell

**Fig. 3 | Phosphoproteomic analysis of LPS-reactive microglia highlights extensive regulation cytoskeletal proteins. a** Experimental outline: primary murine microglia were purified and exposed to LPS for a time course of 30 min to 24 h before lysis, TMT-multiplexing and proteomic and phosphoproteomic analysis. **b** Quantification of significantly up- and downregulated proteins in proteomic analysis of microglia treated with LPS. For full table see Supplementary Data 1. **c**, **d** Gene ontology analysis of significantly upregulated proteins for cellular component and biological process after 8 h LPS treatment. For full table see Supplementary Data 2. **e** Volcano plots of significantly down- and upregulated proteins at indicated timepoints highlighting proteins associated with "Inflammatory response" (green) and selected inflammatory markers (purple). Non-significantly regulated proteins are omitted. GO: Gene oncology. **f** Volcano plot as in e, but

highlighting proteins associated with "Microtubules" or "Microtubule organizing centers" (green) and selected examples (orange). **g** Number of significantly (de-)phosphorylated proteins after LPS treatment at indicated timepoints. For full table see Supplementary Data 1. **h**, **i** Gene ontology analysis of significantly upregulated peptides for cellular component and biological process in the 8 h LPS treatment. For full table see Supplementary Data 2. **j** Volcano plot of de-phosphorylated and phosphorylated peptides after 0.5 h LPS treatment highlighting proteins associated with "Microtubules" (blue) or "Microtubule organizing centers" (green). Non-significantly regulated peptides are omitted. **k** Volcano plot of de-phosphorylated and phosphorylated peptides at indicated timepoints highlighting phosphorylated peptides of Map4 (red) and Stmn1 (teal). Non-significant peptides are omitted. Source data are provided as a Source Data file.

proliferation in cultured microglial cells (Supplementary Fig. 2e). We hypothesized that, in absence of mitotic entry, Cdk1 may play an unexplored role during microglial reactivity and first studied the effects of Cdk1 inhibition with small molecule inhibitors. We could detect attenuation of the LPS-induced increase in cell size after RO-3306 (CDKi) treatment. CDKi treatment also induced a more ramified cell morphology compared to untreated control cells, resulting in an increased cell perimeter despite a reduction in cell size. This effect was dose-dependent and could be replicated with the broader Cdk inhibitor roscovitine (Fig. 6b, c, Supplementary Fig. 5a, b). In addition, we observed attenuation of cell size and ramification phenotypes associated with beta amyloid ant Tau[P301S] fibrils (Supplementary Fig. 5c). These data demonstrate that Cdk1 activation is required for proper reactivity and morphological changes in microglia in vitro.

We next studied the effect of Cdk1 inhibition on centrosomal activation and microtubule polymerization. Signal intensity of gamma-tubulin staining at the centrosome was reduced in reactive microglia treated with CDKi (Fig. 6d). In addition, microtubule pelleting assays showed that CDKi treatment prevented the increase of tubulin polymerization seen in LPS-reactive microglia (Fig. 6e). Tracking MT+TIPs in reactive microglia showed significant reduction of displacement and number of tracks in cells treated with CDKi but their speed and growth direction remained unchanged (Fig. 6f, Supplementary Fig. 5d). Using immunofluorescence, we showed that CDKi treatment in reactive microglia significantly reduced levels of acetylated tubulin and Map4 (Fig. 6g, h). Together these data show that Cdk1 activation is required to induce microtubule polymerization from the centrosomes and their stabilization through posttranslational modifications (PTMs) and MAP binding in LPS-reactive microglia.

To get a better understanding of the pathways of Cdk1-dependent microglial reactivity, we ran an additional proteomic analysis comparing primary microglia treated with LPS, CDKi, or both and found downregulation of 13 and 23 proteins after 8- and 24-h treatments, respectively (Supplementary Data 1, Supplementary Fig. 5e, f). Regulated proteins were enriched in inflammation, cytokine and chemokine release pathways (Supplementary Data 2, Fig. 6i, Supplementary Fig. 5g). Moreover, the downregulation of gamma-tubulin and Stmn1 following LPS stimulation was partially rescued in cells also treated with CDKi, as was the upregulation of inflammatory marker proteins Nos2, Il18 and Ccl7 (Supplementary Data 1, Supplementary Fig. 5h, i). This unbiased proteomics approach confirmed that Cdk1 activity is required for proper inflammatory polarization and microtubule rearrangements in LPS-reactive microglia.

### Cdk1 is the upstream regulator of the microtubule-dependent cytokine trafficking

An important function of microtubule remodeling is to facilitate the active transport of newly synthesized or recycling proteins, such as cytokines, from the Golgi apparatus or endosomal system to the cell periphery where they can be secreted. For example, TNFa is transported to the plasma membrane via endosomal pathways that rely on

transport along microtubules to fuse with the plasma membrane in stimulated macrophages[38]. To determine whether CDKi treatment has an effect on cytokine trafficking and secretion in LPS-reactive microglia, we first stained cells for TNFa. We show that LPS stimulation strongly increased TNFa staining, but this was attenuated in reactive microglia also treated with CDKi. Moreover, TNFa-positive vesicles were clustered more closely in the perinuclear region and penetrated less into the cell periphery (Fig. 6j, Supplementary Fig. 6j). Lastly, we quantified the amount of cytokine release in microglia treated with LPS and/or CDKi and observed a marked reduction of both chemokine and cytokine release in a dose-dependent manner after Cdk1 inhibition (Fig. 6k, Supplementary Fig. 6k). These results clearly demonstrate that Cdk1 activation is not only required for the morphological rearrangements and microtubule remodeling in reactive microglia but also for functional cytokine release. They also stress the importance of the microtubule reorganization in reactive microglia for cytokine transport to the plasma membrane for exocytosis.

### Cdk1 activity is required for the ameboid cell morphology in acute brain slices

Encouraged by the striking results of CDKi on microglial cell morphology in our in vitro experiments, we validated the physiological relevance of Cdk1 activation in brain tissue. Due to the poor in vivo pharmacokinetic properties of the available Cdk1 inhibitors, we were unable to perform in vivo experiments at sufficient and sustained brain-exposure levels and hence assessed the effect of Cdk1 inhibition in an ex-vivo setting. Acute brain slices from Cx3CR1[GFP/wt] mice[39] were exposed to LPS and/or CDKi for 4 h (Fig. 7a). As preparation of acute slices induces inflammatory responses per se, we imaged the center of the slices that still showed intact microglial morphology in control slices. Quantifying the three-dimensional morphology of microglia in this model revealed a decrease in microglial cell volume in LPS-treated slices that could be partially rescued by CDKi (Fig. 7b–d, Supplementary Movie 3). This study showed that Cdk1 activation is required for the morphological changes of microglia in their endogenous environment and confirms the physiological relevance of the pathway.

## Discussion

Reactive microglia cells are important components in many neurological and neurodegenerative diseases and are studied extensively to predict or modify disease outcomes. Yet the molecular processes and mechanisms underlying microglial reactivity have so far rarely been examined. In this study, we propose a model in which initial microglia stimulation results in Cdk1 signaling that in turn drives centrosome maturation, downregulation of Stmn1 and activation of Map4 that together mediate microtubule remodeling, polymerization, and stabilization (Fig. 7e). Microtubule reorganization in LPS-reactive microglia contribute to efficient cytokine secretion by facilitating their transport to the plasma membrane and inducing the morphological changes characteristic of microglial reactivity in vivo.

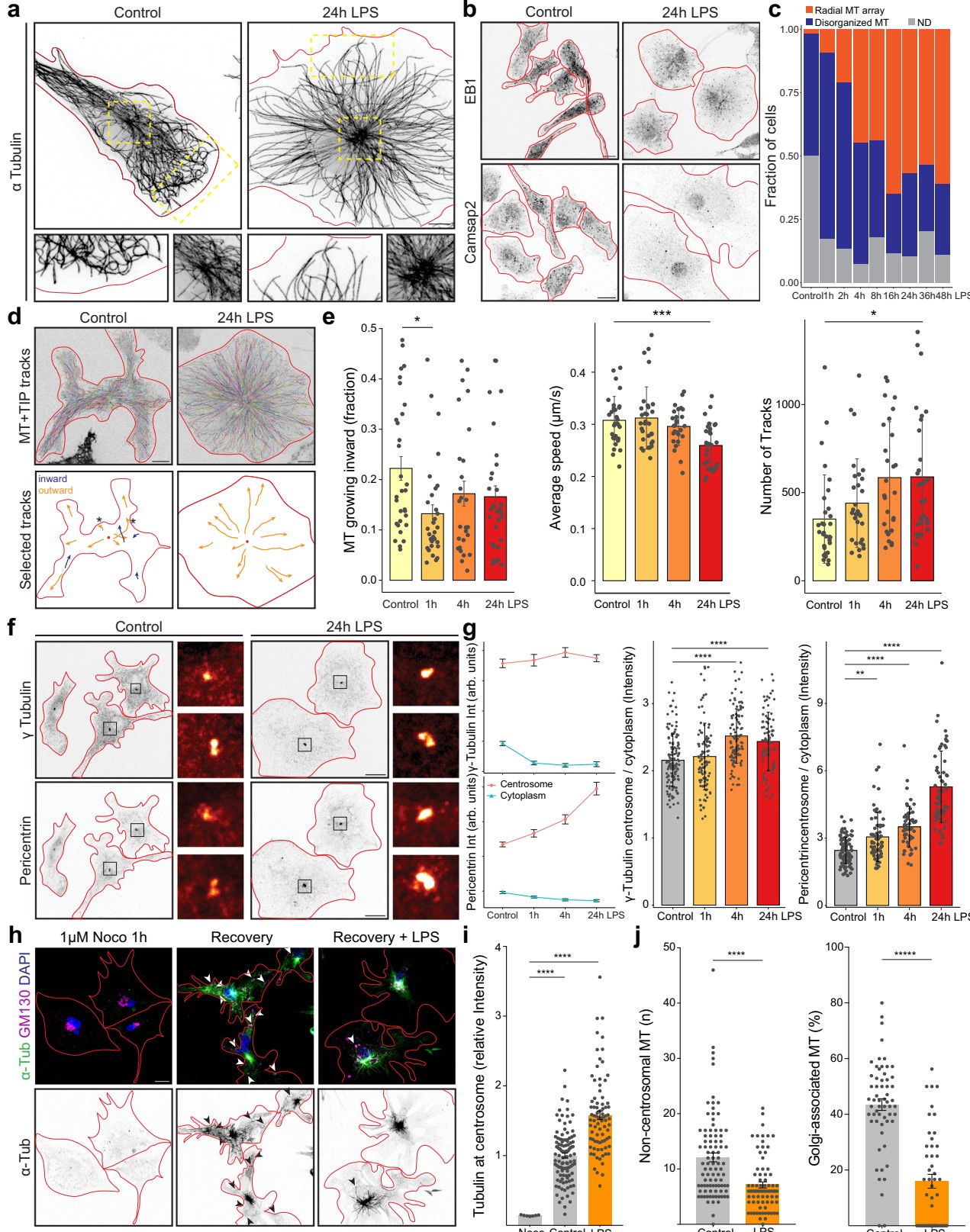

## Polarized microtubule remodeling drives morphological changes in reactive microglia cells

In this study, we uncovered that microtubule remodeling pathways are an early factor that drives the morphological change in LPS-reactive microglia cells. We found that the mechanisms employed by reactive microglia to remodel the cytoskeleton and build the radial microtubule array consist of at least three parts: (1) More microtubules are polymerized from free tubulin by downregulation of the tubulin sequester Stathmin 1. (2) The nucleation site of microtubules is shifted toward the centrosome and reduced at the Golgi apparatus by strong activation of the centrosome and relocation of the gamma-tubulin complex. (3) The resulting microtubule network is stabilized by

**Fig. 4 | LPS-reactive microglia establish a radial microtubule array through centrosome activation and loss of Golgi-associated nucleation. a** Super-resolution immunostaining for alpha-tubulin. Regions in yellow dashes at centrosome and periphery are shown enlarged. **b** Immunostaining for microtubule plus-tip marker EB1 and minus-end marker Camsap2. **c** Quantification of microtubule phenotypes. $n = 58, 76, 76, 67, 84, 43, 58, 54, 54$ cells per time point. **d** Top: MT + TIP tracking in microglia treated with LPS over 5 min. Bottom: a selection of tracks and their direction: inward = blue, outward = orange. Red dot marks position of MTOC. Asterisks indicate tracks turning at plasma membrane. See Supplementary Video 1. **e** Quantification of MT + TIP track direction, speed and number shown in (**d**). $n = 31, 31, 27, 30$ cells from 3 independent cultures. **f** Immunostaining for γ-tubulin and pericentrin. **g** Left: Quantification of γ-tubulin and pericentrin levels at the centrosome and in the cytoplasmic periphery. Right: Quantification of centrosome/cytoplasm ratio. $n = 134, 407, 107, 84$ cells from 3 independent cultures.

**h** Immunostaining for α-tubulin and Golgi marker GM-130 after treatment with nocodazole for 1 h and 1 min recovery in microglia treated with LPS for 24 h. Arrowheads point to microtubules associated with Golgi structures. See Supplementary Video 2. **i** Quantification of α-tubulin intensity at centrosome in microglia treated with or recovered from nocodazole treatment as in (**h**). $n = 7, 109, 82$ cells from 3 independent cultures. **j** Quantification of number of non-centrosomal microtubules and golgi-associated microtubules after recovery from nocodazole treatment. $n = 92, 77,$ (left) 59, 48 (right) cells from 4 independent cultures. Scale bars are 5 μm in a; 10 μm in (**b, d, f, h**). Bars indicate mean ± SE in (**i, j**); mean ± SD in (**e, g**). Statistical significance was calculated with one-way ANOVA and Tukey HSD in (**e, g**), one-way ANOVA and Dunnett's in (**i**), and unpaired, two-sided t-tests in (**j**). Significance intervals p: **** <1e−04 <*** <0.001 <** <0.01 <* <0.05 <ns. Source data are provided as a Source Data file.

recruitment of Map4 and acetylation of microtubules at the MTOC. Together, these parallel pathways build and maintain a radial microtubule array that can facilitate the functional changes discussed below. We also studied the temporal sequence of these events through time course studies both with proteomics and functional assays. The switch from Golgi-mediated microtubule polymerization to centrosomal nucleation is rapid, as evident by MT + TIP tracking, and reached full effect within 1 h after LPS stimulation. Since we observed the continuous accumulation of pericentrin at the centrosome over 24 h, MTOC activation in these cells is both rapid and sustained. The number of MT + TIP tracks measured peaked 4 h after LPS stimulation, at which time cytokine secretion starts to increase dramatically and cell morphology becomes more ameboid. We hypothesize that rapid nucleation of microtubules from the centrosome is key in rearranging them into a radial array to facilitate cytokine secretion. After this initial spurt in microtubule polymerization, the reactive cell enters a new steady state in which it maintains and stabilizes the radial array but decreases new microtubule growth. Our model is consistent with data from bone-marrow-derived macrophages that show centrosomal activation after LPS stimulation as well as recent work in microglial cells[18,40]. However, both studies did not explain the underlying molecular pathways that we uncovered here. Our results may also help explain why Stathmin 1 overexpression reduces macrophage reactivity[41]. Golgi outposts in oligodendrocytes and other brain cell types have been shown to be critical for the maintenance of their morphology and functions[14,42]. It has now been shown that these Golgi outposts are lost in microglia during LPS-reactivity, confirming our observation of loss of golgi derived microtubules[18]. These and our studies also highlight the need to develop in vitro models of microglial cell biology that closely mimic their behavior and morphology in situ and we expect the field to quickly advance with the advent of more sophisticated primary and iPSC-derived microglial cultures[43,44].

## Microtubule remodeling regulates cytokine responses in reactive microglia

We found that microtubule rearrangements in LPS-reactive microglia are required both for the morphological change of cell shape and the release of cytokines. Our data suggest that the dynamics of the microtubule cytoskeleton facilitate microglial reactivity by promoting vesicular transport to and from the plasma membrane[45]. Studies in macrophages have shown that cytokine and Apolipoprotein E secretion occurs from vesicles that need to be transported to the cell surface via microtubules[46–48]. Also release of MMP9 metalloprotease is regulated this way and MMP9-containing vesicles have a preference for transport along stabilized microtubules[49]. Moreover, in macrophages, taxol-induced microtubule stabilization further promoted inflammasome activation and IL-1b release after LPS stimulation[50]. Our findings, that microtubules arrange into stable radial arrays to facilitate these functions, explain how the cytoskeleton influences cellular function. By extending microtubule plus-ends towards the

periphery, kinesin-driven secretory vesicles have direct fast tracks towards the plasma membrane[51,52]. In addition, this model fits with observations that early endocytic vesicles and phagosomes rely on dynein-mediated transport along microtubules of these vesicles to the perinuclear space, where they fuse with lysosomal compartments[53–55]. Intriguingly, the processes of cytokine exocytosis and formation of phagosomes at the plasma membrane can be linked[56], suggesting even higher coordination of vesicular transport in macrophages and microglia. It is clear that any such coordination is strongly facilitated by a uniform radial microtubule array that assigns kinesin- and dynein-driven vesicles a clear subcellular destination. Lastly, our model of microtubule cytoskeleton involvement in microglial reactivity also fits with in vivo studies showing that microtubule-targeting drugs modulate neurodegenerative disease outcomes: Our data demonstrate that inhibiting microtubule dynamics led to a strong inhibition of microglial reactivity and cytokine release in our primary cultures. These findings may help explain why Epothilone D, a microtubule-stabilizing agent, inhibited overall microglial reactivity in a model of Parkinson's Disease[57]. Targeting microtubule dynamics may help untangling the many pathways activated by microglial stimulation and we expect more studies to test their relevance for modulating disease outcome.

## Cdk1 is as an upstream regulator of microtubule remodeling and morphological changes

We found that microtubule stabilization, centrosome nucleation and cytokine trafficking are controlled by upstream activation of Cdk1 kinase in stimulated microglia. We also demonstrated that morphological changes in response to LPS, amyloid beta and Tau[P301S] fibrils were dependent on Cdk1 activation. Since cultured microglia are non-proliferating, there must be parallels between microtubule dynamics in mitosis and microglial reactivity that support a common role for Cdk1 in both processes. During mitosis, the separated centrosomes become the sole nucleators of microtubules, resulting in two radial microtubule clusters. In addition, mitotic microtubules are stabilized upon kinetochore capture. Cdk1 plays a key role in regulating the mitotic spindle and it is conceivable that is has a similar function in stimulated microglia leading to a single radial array in absence of centrosome separation driven by Polo-like kinases during mitosis[58]. Our Cdk1 inhibition experiments demonstrated a dose-dependent effect on cytokine release and morphology changes both in vitro and in situ. This aligns with earlier studies showing Cdk1-dependent Map4 phosphorylation, but it is also conceivable that additional signaling factors play a role in Map4 phosphorylation[59]. Inhibitory phosphorylation of Stmn1 on Ser38 has been shown to be a direct Cdk1 target and regulated by Cdk1 activity previously and is in agreement with our present study[34,35]. In summary, it is intriguing to find an additional non-mitotic role for Cdk1 in the context of microglial stimulation and it will be valuable to determine the effect of Cdk1 inhibition in in vivo neurodegenerative disease models.

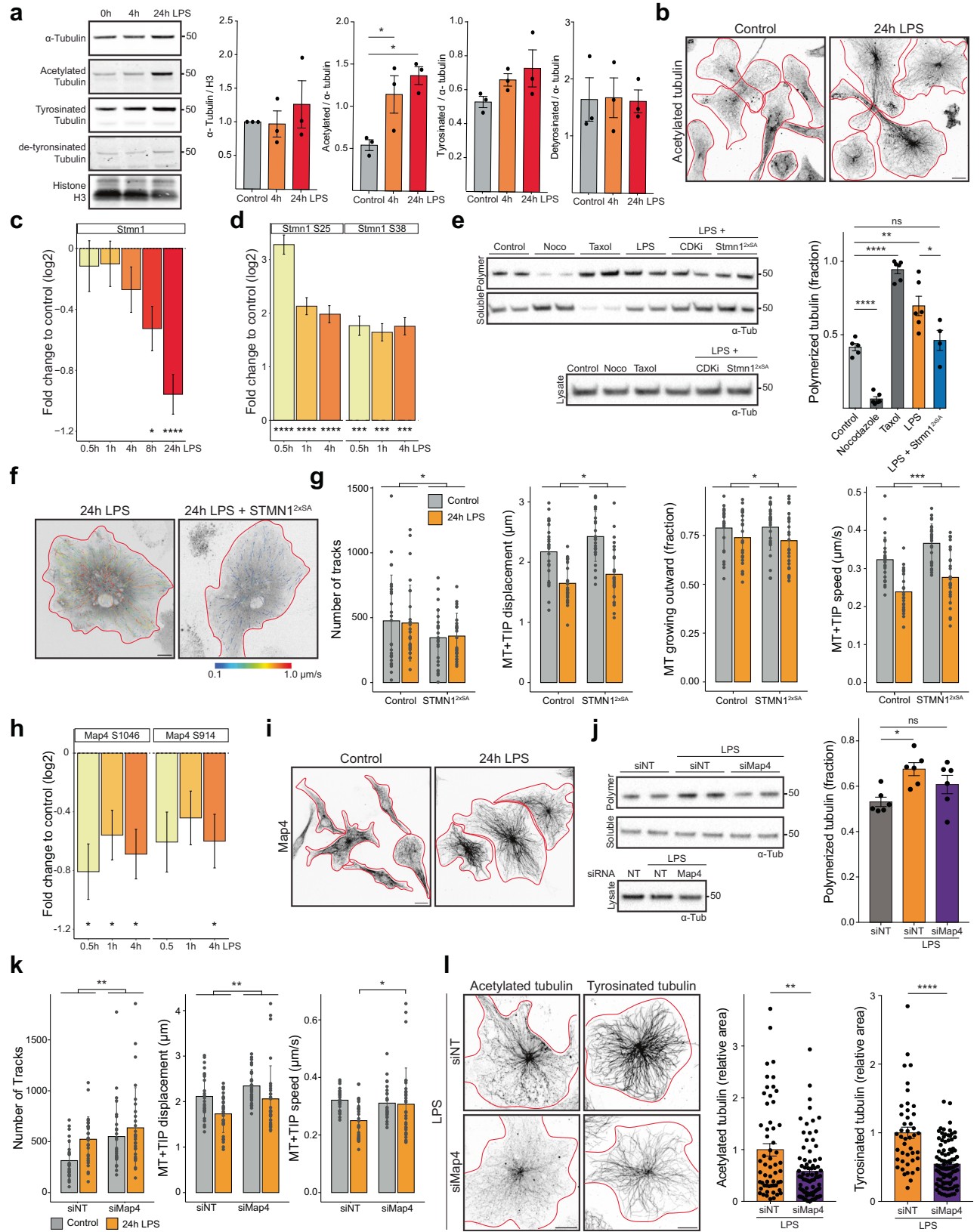

## Methods

### Animals

All animal care and handling procedures were reviewed and approved by the Genentech IACUC and were conducted in full compliance with IACUC policies and the Institute for Lab Animals' guidelines for the humane care and use of laboratory animals.

Animals were housed in specific pathogen-free conditions with 14 h light/10 h dark/day and maintained on regular chow diets and tap water *ad libitum*.

Wild-type C57BL/6N pups were obtained from Charles River, Hollister (CA). C57BL6/J mice were obtained from Jackson Laboratories, Sacramento (CA). Male and female Tau[P301S] mice aged 6 and

**Fig. 5 | Microglial LPS stimulation results in microtubule polymerization and stabilization through Stmn1 and Map4 pathways. a** Representative immunoblot for α-tubulin and posttranslational modifications. Quantification is shown for *n* = 3 independent experiments. **b** Immunostaining for acetylated tubulin.
**c** Quantification of Stmn1 protein levels after LPS treatment compared to control, *n* = 2, 3, 3, 2, 3 replicates. **d** Quantification of Stmn1 phosphorylation on S25 and S38 after LPS treatment, *n* = 2, 3, 3 replicates. **e** Representative immunoblot and quantification of tubulin spin-down assay of microglia treated as indicated or expressing Stmn1^2xSA. Quantification shows fraction of polymerized tubulin for *n* = 5, 6, 4, 6, 6 assays from 3 independent cultures. **f** MT + TIP tracking for 5 min in microglia treated with LPS expressing STMN1^2xSA color coded for their average speed. **g** Quantification of track characteristics in (**f**). *n* = 26, 26, 25, 25 cells from 3 independent cultures. **h** Quantification of Map4 S1046 and S914 phosphorylation,

in MT binding domains, after LPS treatment, *n* = 2, 3, 3 replicates. **i** Immunostaining for Map4. **j** Representative immunoblot and quantification of tubulin spin-down assay of microglia treated with LPS and siMap4, *n* = 6 assays from 3 independent cultures. **k** Quantification MT + TIP track characteristics in microglia treated with siMap4 and LPS. *n* = 28, 31, 25, 30 cells from 4 independent cultures.
**l** Immunostaining for acetylated and tyrosinated tubulin quantification for *n* = 57, 81, 44, 95 cells from 2 independent cultures. Scale bars are 10 μm. Bars indicate mean ± SE in (**a**, **e**, **g**, **j**, **k**, **l**) and mean ± SD in (**c**, **d**, **h**). Statistical significance was calculated with one-sided ANOVA and Dunnett's in (**a**), and Tukey HSD in (**c**, **d**, **e**, **g**, **h**, **j**, **k**) and unpaired, two-sided *t*-tests in (**l**). Significance intervals *p*: **** <1e−04 <*** <0.001 <** <0.01 <* <0.05 <ns. Source data are provided as a Source Data file.

12 months[23,24], female PS2APP x Trem2^KO mice at 6 months[20,60] and male and female Cx3CR1^wt/GFP mice[39] were bred at Genentech.

For the LPS time course study, ~10-week-old C57BL6/J male mice were injected with 1 mg/kg LPS from *Salmonella enterica* (L6143, Sigma-Aldrich, St. Louis, MO) *i.p.* or vehicle control (PBS). Application volume was 10 ml/kg. Animals were monitored and weighted beginning on the day before treatment at timepoints indicated.

All animals were anaesthetized with 2.5% Avertin (2,2,2-tribromoethanol (Sigma-Aldrich); ~0.5 ml/25 g body weight) and transcardially perfused with cold PBS. Blood plasma and brains were collected. One hemibrain was drop fixed in 4% paraformaldehyde (PFA, Electron Microscopy Sciences, Hatfield, PA) for 48 h and processed for immunohistochemistry as described below. The other was stored at −80 °C.

## Embedding, sectioning and immunohistochemistry of brain slices
Harvested brain hemispheres were processed and stained by Neuroscience Associates, Inc (Knoxville, TN USA) according to the following protocol. Hemispheres were treated overnight with 20% glycerol and 2% dimethylsulfoxide to prevent freeze-artifacts. The specimens were then embedded, with 40 brains in the block, arranged for coronal sectioning in a gelatin matrix using MultiBrain® Technology (NeuroScience Associates, Knoxville, TN). After curing with a formaldehyde solution, the block was rapidly frozen by immersion in 2-methylbutane chilled with crushed dry ice and mounted on a freezing stage of an AO 860 microtome. The MultiBrain® block was sectioned coronally with a setting on the microtome of 30 μm. All sections were cut through the entire specimen and collected sequentially into a series of 24 cups. All cups contained Antigen Preserve solution (50-parts PBS pH 7.0, 50-parts ethylene glycol, 1-part polyvinyl pyrrolidone); no sections were discarded.

For immunohistochemistry (IHC), a set of every twelfth section (an interval of 360 microns) for GFAP, Iba1 and CD68, was stained free-floating. All incubation solutions from the primary antibody onward used Tris buffered saline (TBS) with Triton X100 as the vehicle; all rinses were with TBS.

After a hydrogen peroxide treatment and rinses, each set of sections were immunostained with the primary antibodies as shown below, overnight at room temperature. Vehicle solutions contained Triton X100 for permeabilization. Following rinses, a biotinylated secondary antibody (anti IgG of host animal in which the primary antibody was produced) was applied. After further rinses Vector Lab's ABC solution Catalog # PK-6100 (avidin-biotin-HRP complex; details in instruction for VECTASTAIN® Elite ABC, Vector, Burlingame, CA) at a dilution of 1:222 was applied. The sections were again rinsed, then treated with a chromagen: diaminobenzidine tetrahydrochloride (DAB) and hydrogen peroxide to create a visible reaction product. The chromagen for the Iba1 and CD68 stained sections included nickel (II) sulfate as noted below. Following further rinses, the sections were mounted on gelatin coated glass slides, then air dried. The mounted

CD68 slides were counterstained with Neutral Red counterstain. The Iba1-stained slides were dehydrated in alcohols, cleared in xylene and coverslipped with Permount as a bonding medium.

**Iba1 IHC stain:**

| Primary antibody: | Iba1 IHC | Secondary antibody: | Anti-rabbit biotinylated |
|---|---|---|---|
| Source: | Abcam | Source: | Vector |
| Catalog #: | ab178846 | Catalog #: | BA-1000 |
| Host: | Rabbit | Host: | Goat |
| Dilutions: | 75,000 | Dilution: | 1:1000 |
| | | Chromagen: | Ni(II)-DAB |
| | | Color: | Black |

**CD68 IHC stain:**

| Primary antibody: | CD68 IHC | Secondary antibody: | Anti-rabbit biotinylated |
|---|---|---|---|
| Source: | Rockland | Source: | Vector |
| Catalog #: | 600-401-R10 | Catalog #: | BA-1000 |
| Host: | Rabbit | Host: | Goat |
| Dilutions: | 50,000 | Dilution: | 1:1000 |
| | | Chromagen: | Ni(II)-DAB |
| | | Color: | Black |

Slides were scanned at 20× with a Hammamatsu NanoZoomer S60 digital whole slide scanner or Huron Digital Pathologies LE120 scanner.

## Culture of primary murine microglia
Primary murine microglia were harvested from mixed glial cultures by shake-off[14]. Triturated neonatal P2-P3 mouse brains were cultured in DMEM with 10% FBS and cultured for 10–14 days until microglia were loosely attached on top of a layer of astrocytes. Cell culture flasks were then put on a shaker at 300 rpm for 60 min to shake-off microglia from this layer into the supernatant. After washing, microglia were seeded for experiments onto collagen-IV coated plates or glass coverslips and maintained in DMEM with 10% FBS and 10 ng/ml recombinant murine CSF-1 (416-ML, R&D Systems) for up to 4 days.

Cells were treated with 5 μg/ml LPS (00-4976-93, Invitrogen) or 20 ng/ml recombinant murine IL-4 and IL-13 (404-ML, 413-ML), 50 ng/ml TNFa (410-MT), 20 ng/ml IFN gamma (485-MI), 20 ng/ml CCL-2 (479-JE), 20 ng/ml IL-6 (406-ML) or 20 ng/ml IL-10 (417-ML, all R&D Systems). Recombinant human beta amyloid (1–42) monomers (Anaspec Cat No AS-20276) were fibrillized by incubation and agitation at 37 degrees for 24 h, and both added at 110 μM concentration. Recombinant tau wild-type monomers (Human Tau-441 (2N4R), Cat No SPR-479) and tau P301S fibril preparations (Human Tau-441 (2N4R)

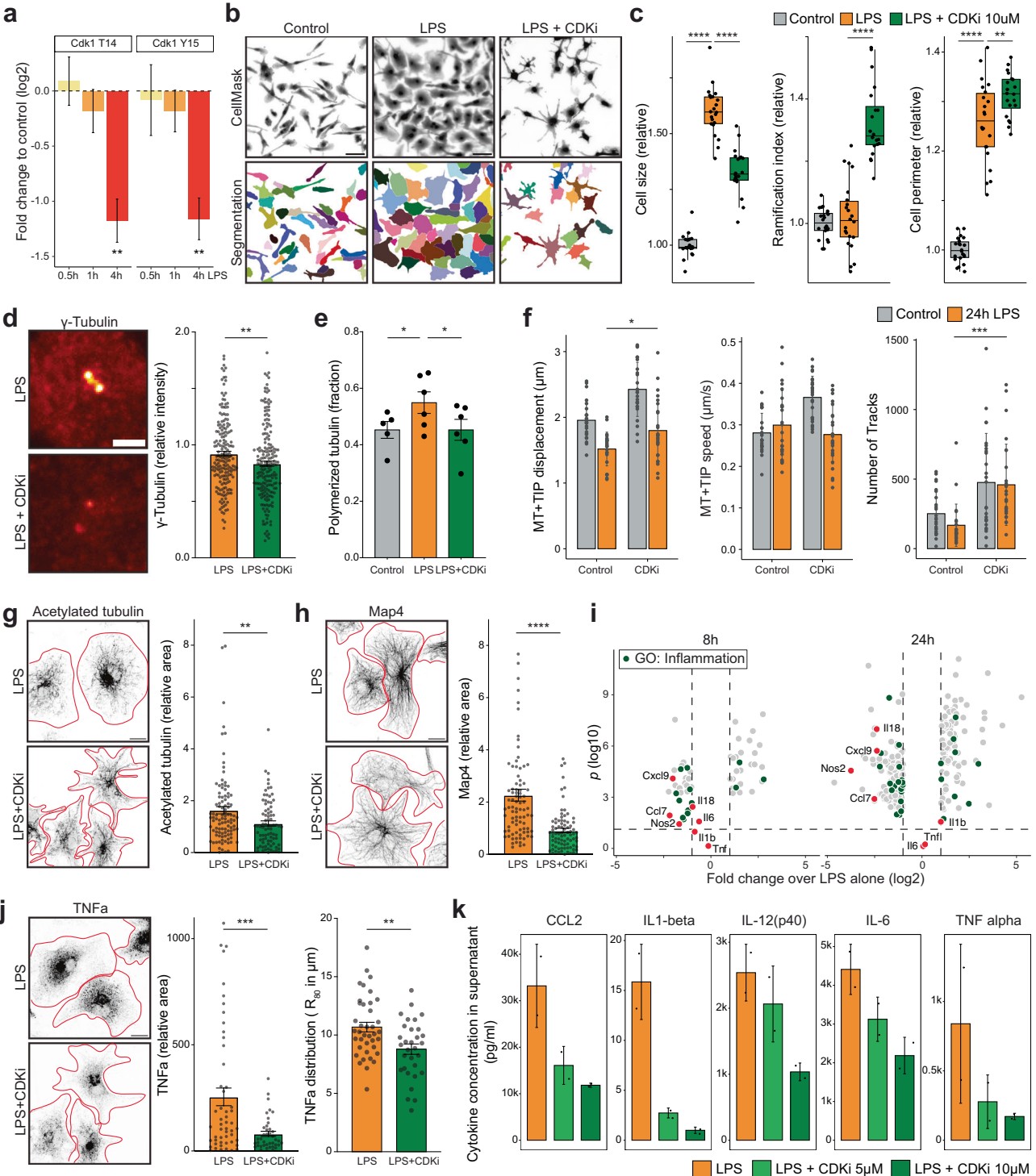

**Fig. 6 | Cdk1 activation is required for microglial reactivity and rearrangement of microtubule cytoskeleton. a** Quantification of Cdk1 T14 and Y15 phosphorylation after LPS treatment, n = 2, 3, 3 replicates. **b** Representative images of CellMask-stained microglia and single-cell segmentation. **c** Quantification of cell size, ramification index and perimeter from (**b**), n = 22, 22, 20 wells from 4 independent cultures. **d** Immunostaining and quantification of γ-tubulin at the centrosome. n = 191, 201 cells from 4 cultures. **e** Quantification of polymerized shown in Fig. 5e. n = 5, 6, 6 assays from 2 independent cultures. **f** Quantification of MT + TIP track characteristics in microglia treated with RO-3306 and LPS. n = 26, 26, 25, 25 cells from 5 independent cultures. **g** Immunostaining and quantification of acetylated tubulin. n = 102, 82 cells from 4 independent cultures. **h** Immunostaining and quantification of Map4. n = 85, 82 cells from 4 independent cultures. **i** Up- and downregulated proteins after 8 and 24 h of LPS and RO3-306 treatment compared

to LPS alone. Proteins associated with "inflammatory response" (green) and selected examples (red) are highlighted. Non-significantly regulated proteins are omitted. **j** Immunostaining and quantification of TNFα. n = 52, 40 cells from 2 independent cultures. Right panel quantifies radius of a circle enclosing 80% of the TNFα signal. n = 39, 31 cells from 2 independent cultures. **k** Quantification of cytokines secretion by microglial cultures, n = 2 replicates. Scale bars are 50 μm in (**b**); 10 μm in (**d, g, h, j**). Bars indicate mean ± SE in (**a, c, d, e, f, g, h, j**); mean ± SD in (**k**). Boxplots show all datapoints, median, 25th and 75th percentile, whiskers are 1.5*IQR. Statistical significance was calculated with one-sided ANOVA and Tukey HSD in (**a, c, e, f**), and unpaired two-tailed t-tests in (**d, g, h, j**). Significance intervals p: **** <1e−04 <*** <0.001 <** <0.01 <* <0.05 <ns. Source data are provided as a Source Data file.

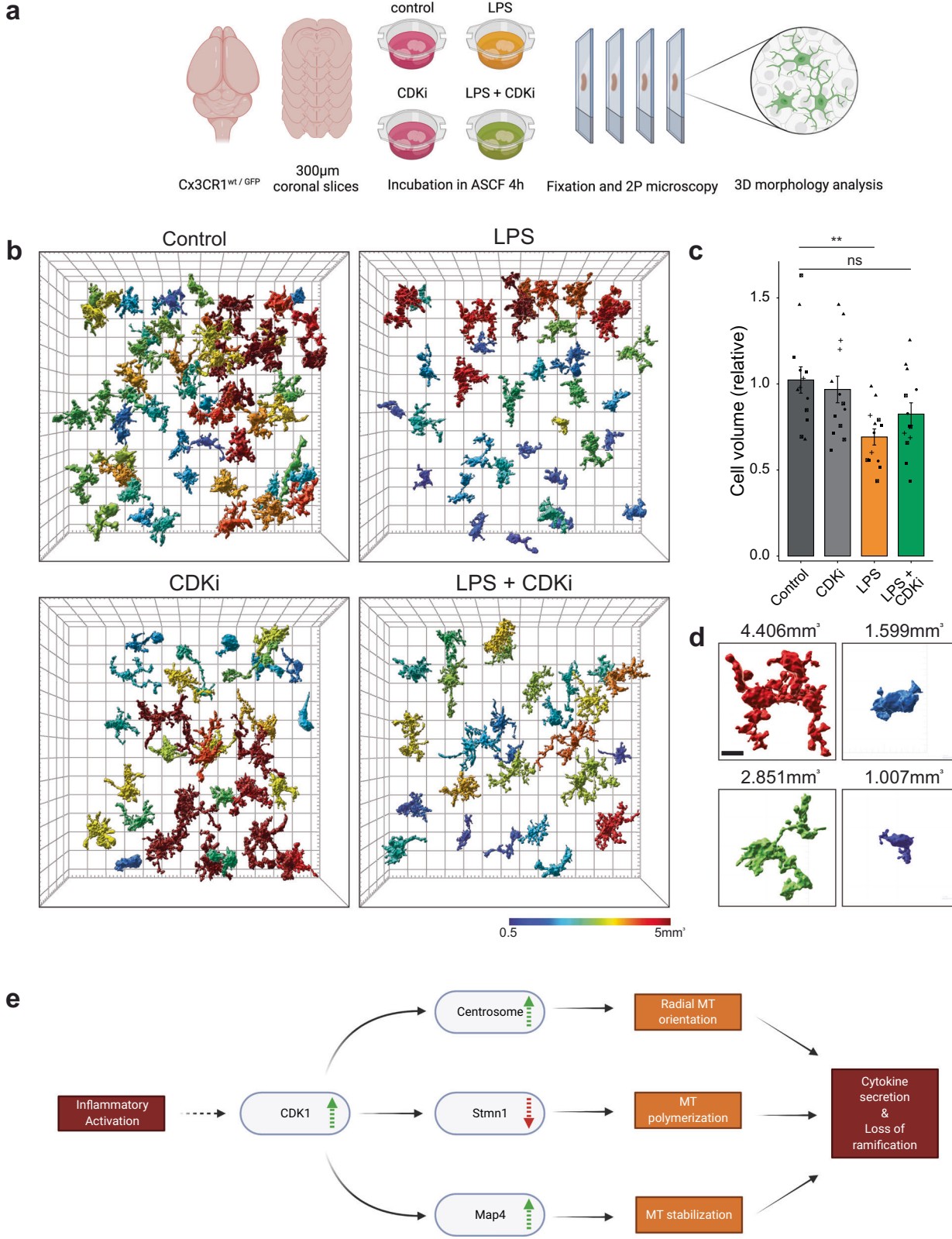

**Fig. 7 | Cdk1 inhibition prevents morphological remodeling of LPS-reactive microglia in ex-vivo brain tissue. a** Experimental outline: 300 μm thick acute slices of Cx3CR1[wt/GFP] brains were incubated in treatments indicated in ASCF for 4 h before fixation and 2-photon microscopy. **b** Representative 3D segmentation result of GFP signal from 150 μm thick image stacks of indicated treatments. Cells are color coded by their volume. Grid spacing is 20 μm in *xy* and 40 μm in *z*. See also Supplementary Video 3. **c** Quantification of cell volumes relative to control condition in acute slices shown in (**b**). Graph shows mean ± SE for *n* = 13 acute sections

recorded in 3 ROIs each from 5 mice. Symbols indicate datapoints originating from the same animals. Statistical significance was calculated with ANOVA and Tukey HSD. Source data are provided as a Source Data file. Significance intervals *p*: **** <1e −04 <*** <0.001 <** <0.01 <* <0.05 <ns. **d** Example 3D segmentation of cells from b with measured volumes indicated. Color-coding as in (**b**). Scale bar is 5 μm. **e** Working model of microglial inflammatory stimulation and resulting MT reorganization as proposed in this paper.

P301S Mutant PFF, Cat No SPR-471, both Stressmarq Biosciences) were added at 4 µg/ml. Nocodazole and taxol (both Sigma) were used at 1 µM unless otherwise noted and RO3306 (CDKi) and Roscovitine / Seliciclib (Selleck Chem) were used at 10 µM or as indicated.

The day after plating, Lipofectamine RNAiMAX (Invitrogen) was used according to manufacturer's instructions to transfect primary microglia with 5pmol siRNA per well on a 24-well plate. We transfected the following pooled on-target Plus siRNA (Dharmacon) for knockdown experiments: siNT (Cat No D-001810-10-20), siMap4 (Cat No L-040500-01-0005) and siAkap9 (Cat No L-041065-01-0005).

For overexpression, we infected microglia with adeno-associated viruses with AAV2/dj serotype and Cbh promoters (Vectorbuilder) at MOI 5E4 for two days. This led to homogenous expression at low levels.

### Single-cell segmentation of Iba1-stained hemibrains and morphological analysis

Regions of interest encompassing medial and lateral cortex and hippocampus were drawn manually on imaged hemibrains. Masked RGB images of Iba-1 DAB staining were converted to 8-bit grayscale images, background subtracted (rolling ball algorithm, radius = 50px) and blurred with a gaussian distribution (sigma = 1). A custom ImageJ macro was written to further segment this image into separate cells. The first step is to segment darker stained cell bodies to identify individual cells and was implemented with the "Area Maxima local maximum detection" function from the SCF Fiji plugin (SCF-MPI-CBG, version 1.2.0). Thresholds for this step were expressed as $Th1 = I_{max} - f_1 * I_{SD}$, where $I_{max}$ and $I_{SD}$ are the maximum and standard deviation of the measured pixel values, respectively, and $f1$ and empirically set factor per dataset. The area threshold was set to 300 px. The resulting areas were then used as seed point for watershed analysis using the "Watershed with seed points" function from the same plugin. Here the threshold was set to $Th2 = I_{mean} + f2 * I_{SD}$, where $I_{mean}$ is the average pixel value in each image and $f2$ set empirically per dataset. Segmented cell area where then filtered to be larger than 900 px and their morphologies measured with MorphoLibJ[61]. The following morphology readouts were used: Ramification index is 1 − circularity, high values indicate many branches; Cell size is the surface area measured in segmentation; Inscribed radius is the radius of the largest disc fitted into the segmented shape (a proxy for cell body size); Feret Diameter is the longest distance between two points on the segmentation boundary; Cell Density is the number of segmented objects per mm²; Geodesic Diameter is the longest path between extremities of the segmented object without crossing its boundary. Measurements were taken from two sections per animal and averaged per animal in all graphs shown.

For morphological analysis of cultured microglia, we followed a similar approach. However, instead of maximum detection, we used a fixed seed dynamics at 10000 in the "H_Watershed" function of the SCF plugin. We set the minimum area size to 200, the threshold to 10× the minimal intensity value per image and turned off object splitting, after empirical testing. Between 5 and 9 technical replicates were imaged per well.

### Immunofluorescence and image analysis

For cell morphology assays, microglia were plated onto 96-well Cell Carrier Ultra plates (Perkin Elmer) at 15k cells per well. Cells were treated as indicated the following day and fixed in 4% paraformaldehyde (PFA) in fixation buffer (150 mM NaCl, 10 mM Hepes, 5 mM EGTA, 5 mM glucose, 5 mM MgCl₂) for 15 min at 37 °C, permeabilized with 0.5% Tx-100 for 5 min and stained with HCS CellMask Far Red (H32721, 1:1000, Molecular Probes) and DAPI for 30 min before imaging on a Leica Thunder widefield microscope with DAPI and Cy5 filter sets at 10x or 20x magnification operated with Leica LAS X software. Seven to nine regions were imaged per biological replicate.

For immunofluorescence, microglia were plated on Collagen-IV-coated glass coverslips in 24-well plates at 80k cells per well in DMEM with 10% FBS and 10 ng/ml recombinant murine CSF-1 (416-ML, R&D Systems). The following day cells were treated as indicated and fixed either with warm 4% PFA in fixation buffer at 37 °C for 20 min or with ice-cold 100% Methanol at −20 °C for 5 min followed by 4% PFA postfixation for 5 min. To extract cytosolic protein in and show microtubule association of Map4, 0.1% Tx-100 was added during the fixation step. Permeabilization was done with 0.5% Tx-100 in PBS and blocking and antibody incubation with 2% bovine serum albumin in PBS. Coverslips were mounted onto glass slides with Prolong Diamond (Thermo Fisher).

Confocal images were acquired on a 3i Marianas spinning disk confocal microscope (Intelligent Imaging Innovations), built on a AxioObserver with 63x Plan Apo 1.4 NA and 100× alpha Plan Apo 1.46 NA objectives (Zeiss) and a CSU-W 50 µm spinning disk (Yokogawa). In addition, the microscope is equipped with a Mesa light homogenizer and 4-line Laserstack with 405, 488, 561 and 640 nm lines (Intelligent Imaging Innovations), Definite Focus 2 (Zeiss), a stage-top incubator (Okolab) and Prime95B cMOS cameras (Photometrics). Data acquisition was done through Slidebook 6 software (Intelligent Imaging Innovations). For super-resolution images a CSU-W SoRa spinning disk (Yokogawa) with ×2.8 magnification lens and 100x objective was used.

Quantification of fluorescent signal intensities and areas was done in ImageJ[62]. Quantification of centrosomal signals was limited to a ROI of 20 × 20 px each compared to three cytoplasmic sites of the same size. The distribution of TNFa-positive vesicles was quantified by measuring the integrated density along a radial profile centered on the MTOC using the "Radial profile" function. The radius from the cell center at 80% of the cumulative integrated density was used as a readout for the distribution of these vesicles.

### Antibodies

We used the following primary antibodies in this study:

| Target | Dilution | Animal | Fixation | Supplier | Catalog# | Lot No |
|---|---|---|---|---|---|---|
| Camsap1l1 (Camsap2) | 500 | Rabbit | PFA | Proteintech | 17880-1-AP | N/A |
| EB1 | 1000 | Mouse | MeOH | BD | 610535 | 6078948 |
| GM-130 | 500 | Rabbit | MeOH | Abcam | ab52649 | N/A |
| Iba1 | 500 | Rabbit | PFA | Wako | 019-19741 | LEJ1842 |
| Pericentrin | 5000 | Rabbit | PFA | Biolegend | 923701 | B215567 |
| Tubulin alpha | 1000 | Mouse | MeOH/PFA | Sigma | T6199 | 029M4842V |
| Tubulin acetylated | 500 | Mouse | PFA | Millipore | T7451 | 206322 |
| Map4 | 500 | Rabbit | MeOH | Abcam | ab245578 | GR3259936-3 |
| Histone H3 (pS28) | 1000 | Rat | PFA | Abcam | ab10543 | N/A |
| Ki67 | 500 | Mouse | PFA | BD | 550609 | N/A |
| Histone H3 | 500 | Rabbit | WB | Cell Signaling | 4499S | 9 |
| Tubulin acetylated | 500 | Rabbit | PFA/WB | Cell Signaling | 5335S | 5 |
| Tubulin detyrosinated | 1000 | Rabbit | WB | Sigma | AB3201 | N/A |

For secondary fluorescent antibodies, we used goat anti-mouse IgG Alexa Fluor Plus 488 (A32723, LOT VA288487) or Alexa Fluor Plus 647 (A32728, LOT UK290265), Alexa Fluor 568 goat anti-mouse IgG (A11031 LOT 2124366), goat anti-rabbit IgG Alexa Fluor Plus 488 (A32731, LOT UK290266) or Alexa Fluor Plus 647 (A32733, LOT UL291628) and Alexa Fluor 568 goat anti-rabbit IgG (A11036, LOT 2155282, all Invitrogen), all at 1:1000 dilution in blocking solution.

For immunoblots we used the following secondary antibodies: Anti-mouse IgG HRP linked (7076, LOT 34) and anti-rabbit IgG HRP linked (7074, LOT 29, both Cell signaling) or IRDye800CW donkey

anti-mouse (926-32212 LOT D21109-15) and IRDye680RD goat anti-rabbit (926-68071, LOT D00819-05, both Licor), all at 1:1000 dilution.

## Quantitative cytokine secretion

Cytokine concentration in cell culture supernatants, plasma and tissue lysate was analyzed using Luminex immuno-multiplexing. All assays were run with Bio-Plex Pro Mouse Cytokine 23-plex Assay #M60009RDPD (Biorad) in technical duplicates for each sample.

## RNA sequencing

Total RNA was isolated using the RNeasy Plus Mini Kit (Qiagen). The concentration of RNA samples was determined by NanoDrop 8000 (Thermo Scientific) and RNA integrity was assessed using the RNA 6000 Nano kit (Agilent). RNA-seq libraries were prepared from 0.5 μg of total RNA using the TruSeq RNA Sample Preparation Kit v2 (Illumina). Libraries were multiplexed and sequenced on an Illumina HiSeq, using a single-end, 50-bp run.

The HTSeqGenie R package (v4.30, https://doi.org/10.18129/B9.bioc.HTSeqGenie) was used to process RNA-seq reads, including filtering, alignment and feature counting. Sequencing reads were mapped to the mouse reference genome (GRCm38) using the GSNAP short read aligner[63]. Only uniquely aligned reads were used for downstream analysis.

## Gene set score analysis

Single-cell RNA sequencing data of the adult murine cortex[64] was imported using the scRNAseq R package (v 2.14, https://doi.org/10.18129/B9.bioc.scRNAseq). For each broad cell type (Microglia, Endothelial cell, Astrocyte, Oligodendrocyte, Oligodendrocyte precursor cell, Glutamatergic neuron and GABA-ergic neuron) the top 10 marker genes, ordered by the Cohen's d, were determined using the scoreMarker function of the scran R package[65].

Raw counts were normalized using the DESeq2 R package[66] and log-transformed as log2(normCount+1). Gene set scores were calculated for each sample as the mean over all marker genes in a top 10 marker gene set.

## qPCR

Primary microglia were seeded into Collagen-IV-coated 24-well plates (Corning) at 130k cells per well in DMEM with 10% FBS and 10 ng/ml recombinant murine CSF-1 (R&D Systems). The following day cells were treated as indicated and RNA was collected with RNeasy Plus Micro kits (Qiagen) according to the manufacturer's instructions. cDNA was generated with iScript cDNA Synthesis Kit (BioRad) followed by qPCR run in triplicate with TaqMan Fast Advanced MasterMix and TaqMan assays in a QuantStudio 7 Flex machine (all ThermoFisher). ddCT quantification was performed in R and Rstudio relative to untreated conditions and Gapdh as housekeeping gene. Each biological replicate was measured in technical triplicates and averaged. We used the following TaqMan assays in this study (all ThermoFisher):

| Target | Assay ID | Dye |
| --- | --- | --- |
| Nos2 | Mm00440502_m1 | FAM |
| Ptgs2 | Mm00478374_m1 | FAM |
| Il1b | Mm00434228_m1 | FAM |
| Arg1 | Mm00475988_m1 | FAM |
| Retnla | Mm00445109_m1 | FAM |
| Chil3 | Mm00657889_mH | FAM |
| Hprt | Mm03024075_m1 | FAM |
| Gapdh | Mm99999915_g1 | VIC |
| Map4 | Mm00485243_m1 | FAM |
| Akap9 | Mm01316486_m1 | FAM |

## Proteomic and phosphoproteomic sample preparation

The two proteomics experiments will be referred to as LPS and LPS+CDKi. For the LPS experiment global proteome, global phosphorylation, and pY profiling were performed. For LPS+CDKi only global proteome profiling was performed.

For both experiments, lysis buffer containing 8 M urea, 150 mM NaCl, 50 mM HEPES (pH 8.0), one complete-mini (EDTA free) protease inhibitor (Roche), and one phosphatase inhibitor (PHOSstop) tablet (Roche) were added to cells in order to lyse the cells. Protein concentrations were then determined by Bradford assay and then disulfide bonds were reduced by incubation with 5 mM DTT (45 min, 37 °C). This was followed by alkylation of cysteine residues by 15 mM IAA (30 min, RT Dark) which was quenched by the addition of 5 mM DTT (15 min, RT Dark). For the LPS+CDKi experiment, proteins were then extracted by chloroform/methanol precipitation and resuspended in digestion buffer (8 M urea, 150 mM NaCl, 50 mM HEPES pH 8.0).

For both experiments, initial protein digestion was performed by the addition of LysC (1:100 enzyme:substrate ratio) followed by incubation at 37 °C for 3 h. Samples were then diluted to 1.5 M urea with 50 mM HEPES (pH 8.0) before the addition of Trypsin (1:50 enzyme to substrate ratio) and incubation overnight at 37 °C. The next day, the resulting peptide mixtures were acidified and desalted via solid phase extraction (SPE; SepPak, Waters). Following this, each sample was resuspended in 200 mM HEPES (pH 8.5) and labeled with tandem mass tags (TMT or TMTpro, Pierce) according to the manufacture instructions. After 1 h of labeling the reaction was quenched by the addition of 5% hydroxylamine and incubated at room temperature for 15 min. Labeled peptides were then mixed, acidified, and purified by SPE.

For the LPS + CDKi experiment TMTpro labeled peptides were separated into 96 fractions by offline basic-reversed phase chromatography before being combined into 24 fraction pools of 12 were analyzed by LC-MS. For the LPS experiment TMT labeled peptides were enriched for phosphotyrosine peptides utilizing the PTMscan pY enrichment protocol (Cell Signaling Technology). Enriched pY peptides were desalted before analysis. Flow-through from the pY enrichment was further enriched for phosphopeptides using TiO₂. Enriched phosphopeptides and the flow-through from TiO₂ enrichment were desalted separately before separation by offline basic-reversed phase chromatography and pooling into 12 phosphopeptide fractions and 24 global proteome fractions, of which 12 were analyzed.

## Proteomic and phosphoproteomic nLC−MS/MS analysis

For nLC−MS/MS analysis, peptides were separated using a Dionex UltiMate 3000 RSLCnano Proflow system (Thermo Fisher Scientific) for the LPS + CDKi experiment or a NanoAcquity UPLC (Waters) for the LPS experiment. For the LPS experiment a gradient of 2% buffer A (98% $H_2O$, 2% ACN with 0.1% formic acid) to 30% buffer B (98% ACN, 2% $H_2O$, 0.1% formic acid) with a flow rate of 450 or 500 nL/min was used to separate peptides over a 25 cm capillary column (100 μm I.D.) packed with Waters nanoAcquity M-Class BEH (1.7 μm) material (New Objective, Woburn, MA). For the LPS+CKi experiment an identical gradient was used but with a flow rate of 300 nL/min and employing an Aurora Series 25 cm × 75μm I.D. column (IonOpticks, Australia).

Samples from the LPS experiment were analyzed using an Orbitrap Fusion Lumos mass spectrometer, while samples from the LPS+CDKi were analyzed using an Orbitrap Eclipse mass spectrometer (Thermo Fisher Scientific, San Jose, CA). For all analyses the SPS-MS3 method was implemented for improved quantitative accuracy[67,68]. For all experiments, intact peptides were surveyed in the Orbitrap ($1 \times 10^6$ AGC, 120,000 resolution) and the top 10 peptides were selected for fragmentation (CID, 30 NCE) and analyzed in the ion trap ($2 \times 10^4$ AGC for LPS and $1.5 \times 10^4$ AGC for LPS + CDKi). Quantitative $MS^3$ scans selected the 8 most abundant fragment ions from the $MS^2$ spectrum

and fragmented them at high energy (HCD, 55 NCE for LPS and 40 NCE for LPS+CDKi) to produce reporter mass ions.

## Proteomic and phosphoproteomic data processing

Assignment of MS/MS spectra was performed using the MASCOT search algorithm to search against all entries for *Mus musculus* (house mouse) in UniProt (downloaded June 2016 for LPS + CDKi or August 2017 for LPS experiments). A search of all tryptic peptides (1 [LPS + CDKi] or 2 [LPS] missed cleavages) was performed and a precursor tolerance of 25 ppm [LPS] or 50 ppm [LPS + CDKi] was used to limit the number of candidate peptides, while a 0.8 Da [LPS] or 0.5 Da [LPS + CDKi] tolerance was used to match MS/MS data collected in the ion trap. Static modifications included TMT on the N-terminus of peptides and lysine residues (+229.16293 for TMT in LPS experiment and +304.2071 for TMTpro in LPS + CDKi experiment) and cysteine alkylation (+57.0215), while variable modifications included methionine oxidation (+15.9949) for all experiments, phosphorylation (+79.9663) for LPS phosphorylation experiments, and TMT labeling of tyrosine (+304.2071) for LPS+CDKi. Peptide spectral matches were filtered to a 2% false discovery rate using a target decoy approach scored with a linear discrimination analysis algorithm before filtering to a 2% false discovery rate at the protein level as previously described[69].

Quantitative values were extracted and corrected for isotopic impurities using Mojave[70]. Additionally, quantitative events with a precursor purity <0.5 or 0.7 (±0.25 Da). The R package MSstatsTMT v.1.6.3 (LPS) or v1.6.6 (LPS + CDKi) was used to preprocess PSM-level quantification before statistical analysis, to have protein quantification and to perform differential abundance analysis. MSstatsTMT estimated log2(fold change) and the standard error by linear mixed effect model for each protein. The inference procedure was adjusted by applying an empirical Bayes shrinkage. To test two-sided null hypothesis of no changes in abundance, the model-based test statistics were compared to the Student *t*-test distribution with the degrees of freedom appropriate for each protein and each dataset. The resulting *P* values were adjusted to control the FDR with the method by Benjamini–Hochberg.

## Gene ontology

Significant hits are defined as proteins with Log2FC > 1 and adjusted *p* values < 0.05. Resulting protein lists were submitted to EnrichR GO analysis through enrichR R package (v3.0, https://maayanlab.cloud/Enrichr/)[71].

Overlays for volcano plots were generated by highlighting proteins associated with GO terms indicated as found on AmiGO[72] (http://amigo.geneontology.org/amigo). Detailed GO outputs for datasets shown are compiled in Supplementary Data 2.

## MT+TIP tracking

Primary microglia were infected with MT + TIP-GFP[73]. Two days after infection cells were imaged on a spinning disk confocal microscope (as above) at 1fps interval for 300 frames and resulting movies were analyzed using TrackMate 7.6[74]. Detection was done with LoG detector at sub-pixel resolution and 0.4 μm radius and threshold varying on expression levels between 1 and 1.8. Spots were linked using the "Simple LAP tracker" (Max linking distance = 1.5 μm, Gap-closing max distance 0.5 μm, Gap-closing max frame gap = 4). Tracks were filtered to have ≥ 4 spots per track and a confinement ratio ≥ 0.8 to exclude non-linear tracks from auto-fluorescent spots. Track results were exported and further summarized per cell and statistically evaluated in R. To determine track directionality, we manually estimated the coordinates of the centrosome as the origin of most tracks in videos and calculated the difference in distance from the first $d_O$ and last $d_n$ spot per track. Tracks with $d_n > d_O$ were defined as outward growing.

## Microtubule spin-down assay

To determine the fraction of polymerized tubulin, we performed spin-down assays using the Microtubule/Tubulin In Vivo Assay Biochem Kit (Cytoskeleton Inc) according to manufacturer's instructions. We over-expressed hSTMN1[S25AS38A]-TagBFP2 (Vectorbuilder) using AAV infection as above.

## Two-photon microscopy of microglia in acute brain slices

Three-month-old male and female CX3CR1$^{wt/GFP}$ mice (*n* = 9) were used for acute slice preparation. After Avertin (250 mg/Kg) induced anesthesia, the animals underwent intracardiac perfusion of ice-cold carbogenated NMDG artificial cerebrospinal fluid (aCSF, in mM): 92 NMDG, 2.5 KCl, 1.25 NaH$_2$PO$_4$, 30 NaHCO$_3$, 20 HEPES, 25 glucose, 2 thiourea, 5 Na-ascorbate, 3 Na-pyruvate, 0.5 CaCl$_2$·2H$_2$O, and 10 MgSO$_4$·7H$_2$O (pH 7.4). Acute brain slices (300 μm) from the forebrain were prepared in the carbogenated NMDG-aCSF at 4 °C. After sectioning, brain slices were first transferred to a recovering chamber containing carbogenated NMDG-aCSF at 37 °C for 15 min and then to a secondary holding chamber containing carbogenated aCSF (mM): 119 NaCl, 2.5 KCl, 2.5 CaCl$_2$, 1.3 MgCl$_2$, 1 NaH$_2$PO$_4$, 26.2 NaHCO$_3$, and 11 glucose (pH 7.4) at 37 °C for another 15 min. The brain slices then cooled down to room temperature and randomly assigned to 4 different incubation chambers containing carbogenated aCSF with the following treatments: vehicle, Drug (RO-3306, 50 μM), LPS (5 ng/ml), LPS + Drug. At 4- and 8-h timepoints after the treatments, brain slices were fixed with 4% paraformaldehyde in phosphate buffer saline (PBS) for 1 h. Then the slices were washed and kept in PBS before mounting. The slices were placed on slides padded with image spacer (SecureSeal, Grace Bio-Labs) and mounted with antifade reagent (ProLong Diamond, Thermo Fisher). Two-photon imaging was performed using a 60× objective (N.A. 1.1) on a two-photon laser scanning microscope (Ultima 2Pplus, Bruker) with a Ti:Sapphire laser (Coherent) at 920 nm to image GFP-positive microglia. Three stacks of 150 μm optical sections (1024 * 1024 pixels) at 1 μm step were imaged from the upper layer of the cortex of each brain slice, starting at the depth of 50 μm from the surface of the slice. Slices from four mice were excluded from analysis because of poor microglial health in the control condition. Volumetric analysis of microglial morphology from 2P microscopy was performed by surface segmentation in Imaris 10.1 (Oxford Instruments).

## Statistics and reproducibility

Data tables were tidied, filtered and summarized in R (version 4.2.3) in RStudio (version 2023.03.0 Build 386) using tidyverse 2.0.0 and statistically evaluated with rstatix 0.7.2 using two-sided *t*-test or one-way ANOVA and Tukey's HSD as indicated. Plots were generated using ggplot2 3.4.2 or ggpubr 0.6.0 with significance intervals at **** <1e−04 <*** <0.001 <** <0.01 <* <0.05 < ns. Alternatively, we used Graph Pad Prism 9 for plotting and statistical analysis with two-sided t-tests and ANOVA with Dunnett's or Tukey's multiple comparison tests as indicated. For statistics in proteomic and RNA sequencing analyses, please see separate "Methods" sections above. All source data for graphs and their statistical analyses are reported in the Source Data. Schematics shown in Figs. 3a, S5e and 7e were created with BioRender (biorender.com).

## Reporting summary

Further information on research design is available in the Nature Portfolio Reporting Summary linked to this article.

## Data availability

The raw proteomic data generated in this study have been deposited in the MassIVE repository under accession code MSV000090254 [https://doi.org/10.25345/C5TT4FZ0P]. The processed proteomics data are available in the Supplementary Information. The RNA

sequencing data generated in this study have been deposited in the NCBI GEO database under accession code GSE238210. Source data are provided with this paper.

## Code availability

Image analysis developed for this paper is based on publicly available functions described in Material & Methods and scripts are available on github [https://github.com/maxadrian/mg-segmentation/].

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

## Acknowledgements

We would like to thank the Genentech vet, necropsy and animal care staff for their support of mouse studies. We thank the Proteomics, Luminex, CALM-EM and Digital Pathology cores at Genentech for technical support, Kimberley Stark for coordination of outsourced histological studies and the Translational Neuroscience group for contributing data from previous mouse studies. We thank the members of the Hoogenraad lab and Christopher J. Bohlen for discussion of this study. We thank Vineet Kulkarni and Xiaoyi Qu for supplying Tau and beta amyloid reagents. PTMScan is a trademark of Cell Signaling Technology. PTMScan studies are performed under a limited use license from Cell Signaling Technology.

## Author contributions

M.A. conceived, performed and analyzed the experiments in this study. M.W. conceived and performed LPS in vivo studies. M.-C.T. performed acute brain sections and 2P microscopy. W.J.M. contributed histological images of microglia in neurodegenerative mouse models and discussed results. O.I.K. and C.G. generated and analyzed RNA sequencing data, respectively. L.P. and T.K.C. performed proteomic analyses under supervision of C.M.R. C.C.H. conceived and supervised the study. M.A. and C.C.H. wrote the manuscript with input from M.W., M.-C.T., C.G., W.J.M. and C.R.

## Competing interests

All authors are employees of Genentech, Inc., a member of the Roche group. The authors declare that they have no additional conflict of interest.
