## [Peer Review File · Nature Communications]

REVIEWER COMMENTS

Reviewer #1 (Remarks to the Author):

The manuscript by Adrian and colleagues aims to study the molecular basis of early morphological changes in microglia reactivity. The manuscript is well written and of high significance, including a wider audience outside the neuroimmunology field (see also comments below). The results are novel and support the authors conclusions - with limitations as noted below. In particular, the authors should be commended on their methods section.

Early microglia responses, in particular those independent of changes in gene expression have so far been largely unstudied. In this context, it is not clear to me why the authors included the in vivo studies in the Alzheimer's disease models? These are clearly not early reactive changes and the overall microglia response in the brains of these mice is confounded by a complex neurodegenerative and neuro-reactive milieu.

The authors also assume that reactive microglia (see also specific comment below in regards to the term 'activated') undergo similar morphological changes in response to stimuli. This is incorrect (e.g. see recent publication by West et al. 2022, J. Neuroinflammation); rather, microglia do show stimulus specific responses which do include clear morphological changes. In light of this, the changes induced by LPS, while an important model, are specific for this stimulus. The manuscript would gain in impact by looking at other stimuli (e.g. do similar changes in morphology and microtubule changes occur in cells stimulated with individual cytokines cytokines?); else, at least a caveat needs to be included throughout the manuscript (incl. title).

Specific comments:

1. I do not see the purpose of the AD mouse model studies. The findings are not novel and the models have very limited correlation with the in vitro model used for all further experiments.
2. The purity of microglia is shown on the basis of Iba1 levels, which itself is a marker that is regulated by inflammation. While I agree that the purity is sufficient for the study, it would be helpful to include additional markers, in particular those that are commonly used to characterise mature microglia (e.g. CSF1R, 4D4, P2RY12, TMEM119, SALL1). Such analysis would help establish that despite the culture conditions, these cells are 'true' microglia. This could be extended to identify the identity of the remaining 3% cells (presumably astrocytes and/or fibroblasts?).
3. Microglia are constantly active; please avoid using the word "activated" or "activation state"; I suggest to use "reactive" or "activity states" instead. Similarly, the term "resting" needs to be avoided.
4. Microglia undergo stimulus-specific morphological changes and the previous assumption of "activated" microglia being amoeboid is just wrong (see also above).
5. Figures should show individual values throughout.
6. It would be good to include more raw data, in particular for the immunoblots (e.g. full gel images)

Reviewer #2 (Remarks to the Author):

In "Polarized microtubule remodeling transforms activated microglia morphology and drives cytokine release," Adrian et al., investigate microtubule remodeling in relation to microglia morphology. The authors first highlight their segmentation model for IBA1+ cells in the steady-state adult mouse hippocampus and cortex, and then show they too can detect changes in morphology with their analysis method in various models of neurodegeneration. The authors further highlight their segmentation model in a primary microglia culture system. The authors then show that hyper-stabilizing and depolymerizing microtubules prevents changes in microglia cell culture morphology following LPS. Then, the authors performed a time course of the proteome and phospho-proteome of

primary microglia stimulated with LPS, finding that in addition to cytokines, proteins involved in the cytoskeleton and microtubule components were increased. In addition to changes in protein levels, there were changes in protein phosphorylation, including in those not changed in number, including Map4 and Stmn1. The authors then investigate the microglia tubulin, finding that with LPS, there is an increase in radially polarized cells. The authors then demonstrate that STMN1, MAP4 and CDK1 play a role in microglia microtubule polarization in vitro. Finally, the authors exposed ex vivo brain slices to LPS and CDKi, showing that inhibition of CDK1 can attenuate LPS induced reductions in cell size.

Overall, there are interesting data presented in the paper, particularly the examination of molecular mediators of microglia microtubule changes. However, aspects of the paper require attention.

1. There is concern about the statistics. Statistics should be performed on the whole mouse/independent cultures, not the individual cells. There are numerous cases where the results come from 2 animals or cultures/group, which is insufficient. A nested statistical approach could be utilized to account for technical replicates from the same biological replicate.
2. The interpretation from culture studies is interesting.
 - a. Morphological changes observed in neurodegeneration in Figure 1 change from highly-ramified to less ramified, could extended ramified processes not travel inward before then radially extending outward? How do the authors know if these are at specific morphologies rather different stages in morphological alterations? Especially given the number of unpolarized cells increases at 1 and 2 hours?
 - b. Do microglia of the same type of polarization (radially versus unpolarized) in the same group (control versus LPS) have the same directionality of MT growth? In other words, do all microglia in LPS grow outward, regardless of morphology? Or do all unpolarized and other cells grow in outward and inward regardless of condition?
 - c. In many cases, the STMN1 mutation (blocking STMN1 phosphorylation) effects seem to be driven by the control group and not due to LPS (or aren't due to LPS at all time points). Similar with the siRNA of Map4. This occurs to a lesser extent in the CDK1 inhibitor studies. However, given similar magnitude changes in the control groups, how can the authors tie these results to inflammation induced changes in these pathways, rather than reducing constitutive/general activity.
 - d. The CDKi proteomics study (unclear if this was done with whole brains or with cultures), the timeline of cytokines examined is curious. The earlier proteomics data shows pro-inflammatory cytokine increases starting at 0.5h. It is unclear based on the presented data if CDK1 leads to increased pro-inflammatory cytokines in response to LPS; although it could potentially contribute to the sustained increases in pro-inflammatory cytokines. (Similar with the supernatant cytokine levels).
3. Segmentation protocol should be better documented and the steps should be visualized in a figure.
4. Overall, there is imprecise language throughout the manuscript:
 - Please avoid the phrasing, "microglia activation," and name specific processes or responses as per field guidelines recommended in Paolicelli et al., 2022, Neuron. See also "resting" microglia.
 - Line 104 – ex vivo or in situ rather than in vivo. All subsequent tissue work should be referred to as such
 - The tone of this manuscript, particularly at the beginning seems to ignore much of the microglial field. This reviewer understands that manuscripts need to "sell" novelty but this should not be at the expense of building on the current field. Segmentation approaches exist. Microglia morphological changes have been demonstrated in the models utilized by the authors. There is interesting data in the manuscript that is underserved by the framing of the first parts of the results. For example, consider, validation of novel morphological workflow in microglia across neurodegenerative disease models rather than, "Morphological microglia changes are a quantifiable hallmark in neurodegenerative models," which implies a novelty to the results when this type of work has been shown before. Lines 100 and 140 also have a similar statement.
 - Line 333, attenuated rather than robust rescue (were comparisons made between the control and CDKi?)

Minor clarifications

- Standard housing conditions are a 12/12 light/dark cycle. Given that there is evidence of circadian changes in microglia, do you think this affected the results? Furthermore, please specify when animals were utilized in their light cycle for experiments.
- Sex of the mice could be better clarified throughout the methods and manuscript.
- Jackson Laboratories, not Jaxon, I believe.

- If the methods could follow the order of the results, that would be helpful to the reader.
- Please include all secondary antibodies used and provide details for them and the DAB IBA1 (dilution, etc) in the corresponding methods sections.
- Imaging details/parameters missing from DAB methods sections
- The authors use broad regions to perform their analyses. Can they confirm that the data compared is from the same "region" i.e. always from the same specific subpart(s) of the hippocampus or cortex? There are noted dorsal/ventral, layer and sub-region differences of microglia that have been extensively reported. For example, Figure 1h-l, representative images are taken from smaller areas of the hippocampus or cortex, but was the entire hippocampus imaged, or just a subset of ROIs? See also Figure 7.
- For the primary cultures, there are known developmental changes to microglia, so do the authors think that the comparison to the adult condition are limited?
- Do the authors have images of IBA1 in addition to alpha-tubulin for the cell culture studies, to map microtubule phenotypes to microglia morphologies (Figures 4 and 5).
- Consider adding scale bar information to images
- Does blocking phosphorylation of STMN1 alter inward dynamics? Similar with MAP4 siRNA and CDKi studies?
- It would have been interesting to see what the various inhibitors of STMN1, MAP4 and CDK1 did to non "activated" microglia. Not including this data reduces the picture of these pathways in regulating inward growing.
- What is normalized cell size used? Especially as it hides the results of the ability of CDKi to reduce LPS induced increases in cell size? More problematic is that using the non-normalized ramification index shows potentially no increases with CDKi compared to controls and an actual reduction in ramification with LPS?
- The authors should consider adding ramification index in addition to cell size, to better compare their ex vivo slice data to the data in Figure 1 and better relate to morphological changes discussed in the paper.
- Keeping scales the same throughout could be helpful for the reader (e.g. Figure 7 c and d, mm3 versus um3).

Reviewer #3 (Remarks to the Author):

In this manuscript by Adrian et al, the authors characterise novel functions of the microtubule cytoskeleton system during the process of microglia activation. They establish a segmentation workflow of Iba-1+ microglia in mouse brain sections in order to quantify morphological changes in different murine neurodegenerative models. From this method development, the authors establish an in vitro model of microglia cell activation involving LPS treatment, which is scrutinised in detail for activation-associated expression of pro-inflammatory markers, for cytokine secretion, for proliferation, as well as for changes in microglia cell size and ramification. Here, first analyses of microtubule function during LPS induced activation on changes in morphology and cytokine secretion is provided. By the use of proteomic profiling of these cultured microglia in the absence and presence of LPS, the authors then identify 'microtubule remodelling pathways' that correlate with distinct changes in microglia cell morphology and organisation of their microtubules following LPS-induced microglia activation in vitro. In vitro and in vivo, the authors suggest CDK-1 as critical upstream regulator of microtubule remodelling and morphological changes.

Overall, the work provides substantial amount of novel data, focussing on a rather understudied but highly relevant topic, the cellular mechanism of microglia activation. While the data provide significant fundamental information, they are highly relevant in the context of inflammation associated with different brain diseases and conditions. However, the authors need to revisit some of their experiments and tone down several of their statements, especially with regard to in vitro data.

- Hypertrophy is widely used as established hallmark of microglia reactivity in vivo. With this in mind, it is not clear if the method presented in Figure 1 clearly advances from previous work using hypertrophy as activity readout. It could have been useful to compare both methods to report on microglia reactivity. That said, the presented data and methods are compelling and very nicely

executed using different neurodegeneration mouse models. A drawback of this figure, however, is the selective use of Iba1 for the identification of microglia. Iba1 is definitely upregulated in reactive microglia, however, many quiescent microglia may well be Iba1- (i.e. Hendrickx et al., 2017; <https://pubmed.ncbi.nlm.nih.gov/28601280/>). Therefore, the authors should include morphometry using additional markers, i.e. identification of TMEM119+/Iba1+/- microglia +/- neuroinflammatory conditions.

- Microglia cultured in FBS rich medium already show some basal activity, but their reactivity can be further enhanced (as shown in the manuscript). Therefore, the graph in S2 b does not show 'zero microglia reactivity'; similarly, the authors need to tone down the text accordingly.

- Figure 2F, G, H: Did the authors analyze cells present in comparable densities? High densities severely affect morphology in microglia. Analyses of low density cultures +/- LPS, should be performed, ideally during live cell imaging before and during LPS incubation.

- I am somewhat puzzled by the description of 'radially polarized' cells. Seems to be an oxymoron. The authors need to comment.

- CDK1 inhibitor RO3306: working concentration of 10 μ M is high. RO3306 blocks CDK1 and other targets already in the nM range, therefore, off target effects are likely

- Most figure legends do not contain sufficient/any information on experimental and statistical parameters

- The title suggests that the paper focusses on the process of cytokine release, which is investigated mainly 'indirectly' through measurement of cytokine secretion into the supernatant. Should be changed to reflect the main findings of the study.

Rebuttal letter: Nature Communications manuscript NCOMMS-22-44776-T ‘Polarized microtubule remodeling transforms the morphology of reactive microglia cells and drives cytokine release’

*** Reviewer #1 ***

The manuscript by Adrian and colleagues aims to study the molecular basis of early morphological changes in microglia reactivity. The manuscript is well written and of high significance, including a wider audience outside the neuroimmunology field (see also comments below). The results are novel and support the authors conclusions - with limitations as noted below. In particular, the authors should be commended on their methods section. Early microglia responses, in particular those independent of changes in gene expression have so far been largely unstudied. In this context, it is not clear to me why the authors included the in vivo studies in the Alzheimer's disease models? These are clearly not early reactive changes and the overall microglia response in the brains of these mice is confounded by a complex neurodegenerative and neuro-reactive milieu. The authors also assume that reactive microglia (see also specific comment below in regards to the term 'activated') undergo similar morphological changes in response to stimuli. This is incorrect (e.g. see recent publication by West et al. 2022, J. Neuroinflammation); rather, microglia do show stimulus specific responses which do include clear morphological changes. In light of this, the changes induced by LPS, while an important model, are specific for this stimulus. The manuscript would gain in impact by looking at other stimuli (e.g. do similar changes in morphology and microtubule changes occur in cells stimulated with individual cytokines?); else, at least a caveat needs to be included throughout the manuscript (incl. title).

We would like to thank the reviewer for the thoughtful comment and truly appreciate that the manuscript was overall well received. We have followed the reviewer's suggestion closely to look at different stimuli, including pathological recombinant proteins (Tau and amyloid beta) and cytokines in-vitro and agree that any stimulus cannot be generalized. However, we did find some similarities in the molecular mechanisms that underlie the morphological rearrangements. We have revised the wording about microglial reactivity throughout the manuscript, de-emphasized the data presented in Fig. 1 and integrated the finding more closely with the in-vitro assays. Please see our response to the specific comments below.

Specific comments:

1. I do not see the purpose of the AD mouse model studies. The findings are not novel and the models have very limited correlation with the in vitro model used for all further experiments.

We agree that the AD models did not connect well with the rest of the findings in the original version of the manuscript. We have reduced the data in this section and emphasized that many of these findings are not novel but used to validate the sensitivity of our segmentation analysis. We also took care to de-emphasize the importance of this section in view of the scope of this manuscript and put it better into context. We have now also included in-vitro assays showing morphological responses by microglia to recombinant Tau and amyloid beta proteins. While the morphological response to each stimulus is different, we were excited to see that Cdk1 inhibition attenuated LPS phenotypes as well as those of Tau and amyloid beta fibrils.

2. The purity of microglia is shown on the basis of Iba1 levels, which itself is a marker that is regulated by inflammation. While I agree that the purity is sufficient for the study, it would be helpful to include additional markers, in particular those that are commonly used to characterize mature microglia (e.g. CSF1R, 4D4, P2RY12, TMEM119, SALL1). Such analysis would help establish that despite the culture conditions, these cells are 'true' microglia. This could be extended to identify the identity of the remaining 3% cells (presumably astrocytes and/or fibroblasts?).

We agree with the reviewer that the characterization of primary microglia with Iba1 staining alone is not ideal. We have added data in Supplementary Fig. 2b showing bulk RNA sequencing for our cultures and demonstrating that the gene set score for microglia markers is highest in these cultures. We did not perform single-cell sequencing to identify the ~3% cells of other origins as this population is very small and would be very difficult to determine. Similar primary microglial cultures have been extensively used in the literature and also characterized by our colleagues before (see e.g. [10.1016/j.cell.2020.07.011](https://doi.org/10.1016/j.cell.2020.07.011)) and we agree with the reviewer that such purity should be more than sufficient for our study.

3. Microglia are constantly active; please avoid using the word "activated" or "activation state"; I suggest to use "reactive" or "activity states" instead. Similarly, the term "resting" needs to be avoided.

We thank the reviewers for pointing this out. We have changed the wording as suggested throughout the manuscript, referring to LPS-reactive and non-stimulated microglia where appropriate. We agree that the previous wording is misleading and did not intend to suggest that non-stimulated microglia were inactive.

4. Microglia undergo stimulus-specific morphological changes and the previous assumption of "activated" microglia being amoeboid is just wrong (see also above).

We thank the reviewer for bringing up this important point. Indeed, reactive microglia have diverging morphological phenotypes depending on the stimulus they encountered, the brain region they reside in, and the time they have been exposed to this signal. We have highlighted this now in the introduction and result sections. What we try to convey in Fig. 1 is that, albeit being exposed to different stimuli, microglia exhibit some general changes in their cell morphology that are shared in different activity paradigms. Our morphological analysis does not dig into the branching patterns where most of the morphological diversity can be observed. Instead, we focused on an overall loss of ramification in-situ and a change in cell size in vitro and were looking for underlying cytoskeletal mechanisms facilitating this change.

In response to the comments above, we have now also tied Fig. 1 and Fig. 2 closer together by performing the morphological analysis in vitro with additional stimuli. We show that indeed morphological phenotypes are different for LPS, recombinant amyloid beta and tau proteins or cytokine stimulation. We are also excited to share that Cdk1 inhibition rescued both the increase in cell size elicited by LPS and amyloid beta fibril reactivity as it did for the decreased cell size in response to Tau^{P301S} fibrils. We believe that this data shows that there are common underlying

molecular mechanisms to these activity states that can contribute to specific responses for each stimulus.

5. Figures should show individual values throughout.

We plotted all quantifications with scatters or as boxplots where possible. One exception is proteomic quantifications where the data had been averaged over the replicates in an upstream pipeline. We do, however, provide all means and SE values in the Source Data and supply the raw proteomic output in a publicly available database for these data. Lastly, for the large Luminex panels in the Supplementary Figures, we did not add the scatter for clarity of the figure panels, but provided the individual values of each replicate measurement in the Source Data.

6. It would be good to include more raw data, in particular for the immunoblots (e.g. full gel images)

We fully agree with the reviewer and we have added all raw measurements underlying all graphs as well as uncropped western blots and proteomic and gene ontology data in the Source Data and Supplemental Data files in the revised version. In addition, we deposited the raw proteomic and RNA sequencing data in public repositories, all in accordance with NPG's editorial policies.

*** Reviewer #2 ***

In “Polarized microtubule remodeling transforms activated microglia morphology and drives cytokine release,” Adrian et al., investigate microtubule remodeling in relation to microglia morphology. The authors first highlight their segmentation model for IBA1+ cells in the steady-state adult mouse hippocampus and cortex, and then show they too can detect changes in morphology with their analysis method in various models of neurodegeneration. The authors further highlight their segmentation model in a primary microglia culture system. The authors then show that hyper-stabilizing and depolymerizing microtubules prevents changes in microglia cell culture morphology following LPS. Then, the authors performed a time course of the proteome and phospho-proteome of primary microglia stimulated with LPS, finding that in addition to cytokines, proteins involved in the cytoskeleton and microtubule components were increased. In addition to changes in protein levels, there were changes in protein phosphorylation, including in those not changed in number, including Map4 and Stmn1. The authors then investigate the microglia tubulin, finding that with LPS, there is an increase in radially polarized cells. The authors then demonstrate that STMN1, MAP4 and CDK1 play a role in microglia microtubule polarization in vitro. Finally, the authors exposed ex vivo brain slices to LPS and CDKi, showing that inhibition of CDK1 can attenuate LPS induced reductions in cell size. Overall, there are interesting data presented in the paper, particularly the examination of molecular mediators of microglia microtubule changes. However, aspects of the paper require attention.

We thank the reviewer for these important comments which we have answered in full below. We are grateful to hear that the reviewer shares our enthusiasm for this study. In our revised version we have addressed the low number of replicates in a number of assays, we clarified and documented our statistical tests in more detail throughout the manuscript and addressed the relevance of Figure 1 to the rest of the study. We especially thank the reviewer for this important feedback on the main text related to Figure 1, which we have extensively edited to address these concerns about undervaluing the field, which was absolutely unintentional. We believe that this has strengthened our manuscript and we detail our responses to your comments below.

1. There is concern about the statistics. Statistics should be performed on the whole mouse/independent cultures, not the individual cells. There are numerous cases where the results come from 2 animals or cultures/groups, which is insufficient. A nested statistical approach could be utilized to account for technical replicates from the same biological replicate.

We thank the reviewer for pointing out these important issues. He/she is absolutely correct and we agree that experimental reproductivity, robustness, and transparency are very important. In response, we repeated all assays to increase the number of independent experiments/cultures and report these numbers throughout the manuscript. In particular, we repeated MT+TIP tracking, immunofluorescence stainings, and acute slice cultures. Throughout, this resulted in more robust statistics but did not change our conclusions and interpretations.

For in-situ data, we show all results averaged per animal, as the reviewer requested.

For acute slices, we increased the N of mice from two to five and averaged individual cell measurements into ROIs and then individual slices. Because brain slices rather than mice were the experimental unit and because a N of five is still not statistically powerful enough to detect changes in subpopulations of cells, we plotted the results and performed the ANOVA on brain slices (2-3 per animal). In the Source Data we also provide the dataset per cell and per slice as well as the output of the Tukey HSD test.

For in-vitro assays pursued two approaches based on the type of experiment performed. For morphological screens, we measured hundreds of cells per well and averaged them per well, the experimental unit that was manipulated. For each of these experiments, we identify the number of wells and the number of independent cultures that these cells came from. For experiments, where we measure detailed immunofluorescence stainings or image the dynamics of individual microtubule plus-ends, we report the results per cell rather than averages per cultures. Variation between primary cells in culture is typically larger than batch-to-batch culture differences and we would like to show the spread between individual cells to show robustness of the assay. We provide all raw measurements along with their detailed statistical test outputs from R or Prism in the Source Data and identify cell and culture numbers in the figure legends throughout.

2. The interpretation from culture studies is interesting. a. Morphological changes observed in neurodegeneration in Figure 1 change from highly-ramified to less ramified, could extended ramified processes not travel inward before then radially extending outward? How do the authors know if these are at specific morphologies rather different stages in morphological alterations? Especially given the number of unpolarized cells increases at 1 and 2 hours?

Terrific point - it is very much possible that the cytoplasm and membrane pools 'gained' by retracting the protrusions contribute to the process of growing the cell body. We believe the two processes are linked. In situ, the growth of cell bodies (Supplementary Fig. 1c - Inscribed radius - a measure for cell bodies without protrusions) is typically correlated with smaller overall cell size (a measure taking into account cell bodies and protrusions).

In vitro, our primary cells are already less ramified in their non-stimulated condition. Hence in these assays, we use cell size as a measure to describe the overall flattening and growth of the cells on tissue-culture substrates.

In both cases, we believe that the underlying remodeling of microtubules extending from the centrosome leads to an increase in cell (body) size that requires the cell's protrusions to shrink. We, unfortunately, do not know how this process is maintained or changed over longer time ranges in late pathological conditions in vivo.

2b. Do microglia of the same type of polarization (radially versus unpolarized) in the same group (control versus LPS) have the same directionality of MT growth? In other words, do all microglia in LPS grow outward, regardless of morphology? Or do all unpolarized and other cells grow in outward and inward regardless of condition?

We thank the reviewer for this interesting question. Indeed, cell morphology can be heterogeneous in both control and LPS-stimulated conditions. We did our best to image cells in all conditions without bias and record MT+TIP dynamics from a representative group of cells per treatment condition. Especially at early time points of LPS treatment, the cell morphology

can vary substantially between cells. As quantified in Figure 4e, you can appreciate that a subset of cells still has a fraction inward-growing microtubules and these are indeed seen mostly in less radially-polarized cells. In this assay, we did not express any membrane proteins that could allow us to accurately characterize the cell morphology of each cell we imaged and therefore did not further quantify this correlation.

2c. The CDKi proteomics study (unclear if this was done with whole brains or with cultures), the timeline of cytokines examined is curious. The earlier proteomics data shows pro-inflammatory cytokine increases starting at 0.5h. It is unclear based on the presented data if CDK1 leads to increased pro-inflammatory cytokines in response to LPS; although it could potentially contribute to the sustained increases in pro-inflammatory cytokines. (Similar with the supernatant cytokine levels).

The CDKi proteomics study in Fig. 6 was performed on primary microglia cultures, just as the LPS time course study in Fig. 3. We show the experimental setup in Supplementary Fig. 5e. The CDKi study did not include phospho-proteomic analysis, hence we chose to focus on timepoints at which we would expect protein translation levels to change, 8 and 24h, that we also used in the LPS proteomic study. The 30 min to 4 h time points in the initial study were mainly chosen for phospho-proteomic analysis, as changes in translation and protein abundance usually take several hours to reach maximum effect. Indeed, several cytokines can be picked up earlier as shown in Fig. 3 but protein levels of these cytokines remained elevated at 8 and 24 h. We agree that the experiment does not address whether CDKi inhibits the onset or sustained release of cytokines. However, the Luminex assay in Fig 5k shows the cumulative amount of cytokines released over 24h.

3. Segmentation protocol should be better documented and the steps should be visualized in a figure.

We have mentioned all steps of both segmentation protocols in the Materials and Methods section and also now added a panel summarizing the steps visually in Supplementary Figure 1a.

4. Overall, there is imprecise language throughout the manuscript:
• Please avoid the phrasing, “microglia activation,” and name specific processes or responses as per field guidelines recommended in Paolicelli et al., 2022, Neuron. See also “resting” microglia.

We thank the reviewer for addressing these inaccuracies. We have changed the language about microglial reactivity and stimulations throughout the manuscript and believe that this better and more accurately reflects our experiments now.

• Line 104 – ex vivo or in situ rather than in vivo. All subsequent tissue work should be referred to as such

We have now clearly distinguished between in vitro in-situ, acute slices and in-vivo processes throughout the manuscript.

- The tone of this manuscript, particularly at the beginning seems to ignore much of the microglial field. This reviewer understands that manuscripts need to “sell” novelty but this should not be at the expense of building on the current field. Segmentation approaches exist. Microglia morphological changes have been demonstrated in the models utilized by the authors. There is interesting data in the manuscript that is underserved by the framing of the first parts of the results. For example, consider, validation of novel morphological workflow in microglia across neurodegenerative disease models rather than, “Morphological microglia changes are a quantifiable hallmark in neurodegenerative models,” which implies a novelty to the results when this type of work has been shown before. Lines 100 and 140 also have a similar statement.

We thank the reviewer for this candid feedback. It was not our intention to oversell our morphological analysis. Indeed, there are many algorithms identifying microglial morphologies at different levels of detail and technical sophistication, several of which we mention in this section. We have rewritten this section extensively to remove any suggestion of inventing a first-of-kind analysis. The strength of our approach lies in its relative simplicity with enough sensitivity to detect changes in cell morphology of common Iba1-DAB stainings. We also took care to de-emphasize the importance of this section in view of the scope of this manuscript and connect it more directly to our in-vitro data by adding additional stimulations to our panel in Fig. 2a.

Minor clarifications

- Line 333, attenuated rather than robust rescue (were comparisons made between the control and CDKi?)

We thank the reviewer for this suggestion and have altered the text accordingly.

- Standard housing conditions are a 12/12 light/dark cycle. Given that there is evidence of circadian changes in microglia, do you think this affected the results? Furthermore, please specify when animals were utilized in their light cycle for experiments.

Our mouse facilities run on a 14/10 hour light/dark cycle as described in the methods section. Since we do not shift the circadian rhythm of the animals, we do not believe there to be an effect of that. All mice were handled at the same time points for injections (mornings, ~1 h post light on). Mice sacrificed for primary cultures were always collected at mid day, 2 days after arrival in their holding rooms.

- Sex of the mice could be better clarified throughout the methods and manuscript.

Sex of mice is now indicated clearly in the methods section for each strain. We did not expect to see differences between sexes based on literature (10.1002/glia.24427) and did not in studies containing both males and females. The sex of each mouse is indicated in the Source Data tables for panels in Fig. 1.

- Jackson Laboratories, not Jaxon, I believe.

We thank the reviewer for this correction and have altered the text accordingly.

- If the methods could follow the order of the results, that would be helpful to the reader.

We thank the reviewer for this suggestion and have altered the text accordingly.

- Please include all secondary antibodies used and provide details for them and the DAB IBA1 (dilution, etc) in the corresponding methods sections.

We have added product details and catalog numbers for secondary antibodies in the revised manuscript. We also added a complete protocol for the Iba1 histochemistry performed by our CRO.

- Imaging details/parameters missing from DAB methods sections

We now added a complete protocol for the Iba1 histochemistry performed by our CRO in the methods section.

- The authors use broad regions to perform their analyses. Can they confirm that the data compared is from the same “region” i.e. always from the same specific subpart(s) of the hippocampus or cortex? There are noted dorsal/ventral, layer and sub-region differences of microglia that have been extensively reported. For example, Figure 1h-l, representative images are taken from smaller areas of the hippocampus or cortex, but was the entire hippocampus imaged, or just a subset of ROIs? See also Figure 7.

Yes, all regions in Fig. 1 were quantified as shown in Fig. 1a. All ROIs were manually curated to encompass similar cortical and hippocampal regions at similar sectioning depth. By eye, the cortical regions appeared very homogenous in control mice while the layering of the hippocampus could be appreciated. We would also like to stress that all quantifications in Fig. 1 were carried out in the entire regions marked by red boxes in Fig. 1a and not on the zoomed images. We measured thousands of cells per ROI that were subsequently averaged per slice and animal. We have highlighted this now in the updated figure legend. In Fig. 7, we show the whole field of view that we analyzed in panel b. For all of these quantifications, we now also supply all raw measurements in the Source Data.

- For the primary cultures, there are known developmental changes to microglia, so do the authors think that the comparison to the adult condition are limited?

We thank the reviewer for bringing up this important topic. Absolutely, we agree that cultured primary microglia from mouse pups are indeed different from adult cells in situ. We addressed these differences in the text. Importantly, as mentioned in the discussion, our observations in the primary cells fit with data known from other in vitro and in vivo studies. We believe that the power of molecular manipulations in primary cells is important to study the underlying mechanisms that cannot be easily addressed in situ even if they are limited by mimicking only certain aspects of microglial morphology and behavior. We hope that future research will bridge

this gap by developing more refined culture methods that better reflect the cells' environment in situ as we mention in the Discussion.

• Do the authors have images of IBA1 in addition to alpha-tubulin for the cell culture studies, to map microtubule phenotypes to microglia morphologies (Figures 4 and 5).

We outlined the cell morphology based on unspecific binding of the antibody for illustrative purposes in the figure but do not have a specific staining that we can use for quantitation. Unfortunately, we cannot generate a direct comparison of cell shape and microtubule staining with our datasets.

• Consider adding scale bar information to images

We provide scale bar information for all microscopy images in the figure legends and have standardized their sizes as much as possible throughout this manuscript.

• Does blocking phosphorylation of STMN1 alter inward dynamics? Similar with MAP4 siRNA and CDKi studies?

siMap4 treatment, Stmn1^{2xSA} overexpression and CDKi treatment does not alter the microtubule growth direction significantly. We have added new data to all MT+TIP tracking experiments and updated the panels in main and supplementary figures. We believe that this effect is mainly due to centrosomal activation that we would only expect to be altered by CDKi. We have not investigated if Cdk-1 inhibition has an effect on Golgi-localized microtubule nucleation. This is the main driver for inward growing microtubules in non-stimulated microglia.

• It would have been interesting to see what the various inhibitors of STMN1, MAP4 and CDK1 did to non “activated” microglia. Not including this data reduces the picture of these pathways in regulating inward growing.

We agree with the reviewer that this is an interesting question. We had addressed this partially in the initial version in key experiments, like the cell morphology assays (Supplementary Fig 5a,b) and in the acute slices experiment (Fig7 a,b). We now also show non-LPS stimulated data in other assays, e.g. in MT+TIP assays. In addition we now express the immunofluorescence levels in Fig 6 relative to non-stimulated control cells (Fig6 d,g,h,j).

• What is normalized cell size used? Especially as it hides the results of the ability of CDKi to reduce LPS induced increases in cell size? More problematic is that using the non-normalized ramification index shows potentially no increases with CDKi compared to controls and an actual reduction in ramification with LPS?

We used normalization to compare cell morphologies between experiments. As batches of primary cells differ, we set the average of the control condition to 1. This still allows to see the spread of the datapoints within each group and to examine the effect of treatments on the cell size and more conveniently shows the effect size of the various experimental conditions. This metric alleviates some batch effects that affected overall cell size of a given culture.

- The authors should consider adding ramification index in addition to cell size, to better compare their ex vivo slice data to the data in Figure 1 and better relate to morphological changes discussed in the paper.

We agree with the reviewer that a combination of these measures gives the best description of the morphology phenotype. We now show ramification index and cell size for all experiments either in primary or supplemental figures. As we mention in the main text, we do find that in vitro, cell size is the most sensitive measure as primary microglia in monocultures are often not very ramified, but elongated and small, in non-stimulated conditions. We mention this limitation in the results and discussion sections and hope that the field will establish better culture conditions for microglial cells to mimic cell ramification more accurately in vitro.

- Keeping scales the same throughout could be helpful for the reader (e.g. Figure 7 c and d, mm³ versus um³).

We thank the reviewer for this suggestion and have changed the figure accordingly.

*** Reviewer #3 ***

In this manuscript by Adrian et al, the authors characterise novel functions of the microtubule cytoskeleton system during the process of microglia activation. They establish a segmentation workflow of Iba-1+ microglia in mouse brain sections in order to quantify morphological changes in different murine neurodegenerative models. From this method development, the authors establish an in vitro model of microglia cell activation involving LPS treatment, which is scrutinised in detail for activation-associated expression of pro-inflammatory markers, for cytokine secretion, for proliferation, as well as for changes in microglia cell size and ramification. Here, first analyses of microtubule function during LPS induced activation on changes in morphology and cytokine secretion is provided. By the use of proteomic profiling of these cultured microglia in the absence and presence of LPS, the authors then identify 'microtubule remodelling pathways' that correlate with distinct changes in microglia cell morphology and organisation of their microtubules following LPS-induced microglia activation in vitro. In vitro and in vivo, the authors suggest CDK-1 as critical upstream regulator of microtubule remodelling and morphological changes. Overall, the work provides substantial amount of novel data, focussing on a rather understudied but highly relevant topic, the cellular mechanism of microglia activation. While the data provide significant fundamental information, they are highly relevant in the context of inflammation associated with different brain diseases and conditions. However, the authors need to revisit some of their experiments and tone down several of their statements, especially with regard to in vitro data.

We would like to thank the reviewer for these very thoughtful comments on our manuscript and appreciate that our findings were well received. We have taken great care to address your concerns in our revised version. We validated our segmentation approach with Tmem119 staining in slices adjacent to Iba1 staining, revised wording around microglial activity states and increased statistical transparency throughout the manuscript. Please find our detailed responses to your comment below.

- Hypertrophy is widely used as established hallmark of microglia reactivity in vivo. With this in mind, it is not clear if the method presented in Figure 1 clearly advances from previous work using hypertrophy as activity readout. It could have been useful to compare both methods to report on microglia reactivity. That said, the presented data and methods are compelling and very nicely executed using different neurodegeneration mouse models. A drawback of this figure, however, is the selective use of Iba1 for the identification of microglia. Iba1 is definitely upregulated in reactive microglia, however, many quiescent microglia may be Iba1- (i.e. Hendrickx et al., 2017; <https://pubmed.ncbi.nlm.nih.gov/28601280/>). Therefore, the authors should include morphometry using additional markers, i.e. identification of TMEM119+/Iba1+/- microglia +/- neuroinflammatory conditions.

Indeed Figure 1 confirms that microglia react in known models of neurodegeneration. Here we demonstrate that our analysis pipeline picks up these changes in cell morphology and allows us to quantify the phenotypes reliably. There are several useful algorithms described in the literature that analyze microglial morphology with various approaches and levels of detail. Our approach is not meant to be superior to them, rather it is a convenient and purpose-built way to

analyze widely available Iba1-DAB-stained tissue. We agree that benchmarking sensitivity of various approaches under standardized conditions for several microglial markers would be an interesting and worthwhile endeavor. However, such comparison would require a ground truth annotation that is not feasible to perform on thousands of cells per ROI and divert strongly from the scope of the current manuscript. We have described our approach in detail and would gladly contribute to such an effort in a follow-up study.

We developed this algorithm specifically for Iba1-DAB staining as this protocol results in darkly stained cell bodies and lighter protrusions. This helps us to identify individual cells in a first step and then lets us use watershedding to find cell borders between their connected protrusions. We would also like to point out that Iba1 cell density, under this staining protocol, does not strongly change in the neurodegenerative mouse models (Supplementary Fig 1a). We agree that we might miss certain microglial populations by only looking at Iba1 staining. However, even in this population a clear morphological response can be measured between non-stimulated and stimulated microglia. In vitro, we used a cell mask staining that stains all cells irrespective of Iba1 expression.

Nevertheless, we agree that Iba1 is regulated during microglial reactivity. We have now applied our algorithm to wildtype mice stained for Iba1 and Tmem119 in adjacent slices and found no statistical differences in cell number and ramification index (Supplementary Fig 1b). Detection of Tmem119 with this particular algorithm was hard to parameterize with the given thresholds, as cell bodies can be missed and we had to substantially change the threshold parameters. We mention this limitation in the main text.

- Microglia cultured in FBS rich medium already show some basal activity, but their reactivity can be further enhanced (as shown in the manuscript). Therefore, the graph in S2 b does not show 'zero microglia reactivity'; similarly, the authors need to tone down the text accordingly.

We unreservedly agree with the reviewer that no microglial population has 'zero activity' - especially in culture - and we did not mean to suggest this with this panel. We have rephrased the passage in the text as we aim to demonstrate that primary microglia can be further polarized into more inflammatory or more anti-inflammatory pathways from their basal activity in vitro. This experiment is meant to highlight that these cultured microglial cells have the capacity to react to strong stimuli with specific transcriptional responses, not that unstimulated cells are inactive. In addition, we have changed the language throughout the manuscript to better reflect the specific stimulations used in the experimental assays, rather than referring to 'activated' microglia.

- Figure 2F, G, H: Did the authors analyze cells present in comparable densities? High densities severely affect morphology in microglia. Analyses of low density cultures +/- LPS, should be performed, ideally during live cell imaging before and during LPS incubation.

We agree with the reviewer that cell density has a strong effect on microglial reactivity. Before starting the assays described here, we standardized seeding densities and kept their numbers comparatively low and constant. We listed the seeding densities in the methods section for

reference. It should also be noted that in our culture conditions, we did not see cell proliferation, keeping the cell numbers constant throughout the course of these experiments (1-3 days).

- I am somewhat puzzled by the description of ‘radially polarized’ cells. Seems to be an oxymoron. The authors need to comment.

We thank the reviewer for pointing out this inaccuracy. We distinguish two microtubule phenotypes in our cultures. In non-stimulated microglia, we tend to see more cells with dense, tangled and disorganized microtubule networks. In LPS-stimulated microglia we mostly see radial microtubules arrays emanating from the MTOC towards the cell periphery. We have now clarified the language in the text to clarify that we are referring to radial microtubules rather cell polarization.

- CDK1 inhibitor RO3306: working concentration of 10 μ M is high. RO3306 blocks CDK1 and other targets already in the nM range, therefore, off target effects are likely

We thank the reviewer for addressing this important topic. The reviewer is correct that lower concentrations of RO-3306 are sufficient in in vitro experiments with purified kinases. However, for cell-based assays, the standard in the literature is 10 μ M. Many studies on mitotic cell signaling use 10 μ M RO-3306 to inhibit Cdk1 in cell culture. In addition, we titrated RO-3306 and roscovitine in our assay in Supplementary Fig5a, b. We see a dose dependent attenuation of the LPS phenotype starting at 3 μ M RO-3306. We also see a dose-dependent effect of RO-3306 on cytokine secretion in Fig. 6k and Supplementary Fig. 5k that partially inhibits release at 5 μ M. We do agree that follow-up studies may be needed to test the dependence on microglia on Cdk-1 signaling genetically. Since Cdk-1 is an essential gene, we would suggest studying this in a conditional Cdk-1 KO restricted to microglial or macrophage populations.

- Most figure legends do not contain sufficient/any information on experimental and statistical parameters

We agree with the reviewer that transparency about statistical testing is very important. We added n, and variance identifiers to all figure legends consistently. We now also state the type of statistical comparison in the figure legends. For simplicity, we show p-value groups with stars in the figure panels but include exact p-values and statistical output from R and Graphpad into the Source Data file for each individual panel. We also expanded the Methods section on statistical analysis.

- The title suggests that the paper focuses on the process of cytokine release, which is investigated mainly ‘indirectly’ through measurement of cytokine secretion into the supernatant. Should be changed to reflect the main findings of the study.

We would like to point the reviewer to our data on TNFa transport and distribution in microglia cells (Fig. 6j, Supplementary Fig. 6j). We show that LPS stimulation strongly increased TNFa staining, but this was attenuated in reactive microglia also treated with CDKi. Moreover, TNFa-positive vesicles were clustered more closely in the perinuclear region and penetrated less into the cell periphery. In addition to the Luminex cytokine secretion assay results, these data stress

the importance of the microtubule-dependent re-organization in reactive microglia for cytokine transport, such as TNF α to the plasma membrane for exocytosis.

REVIEWERS' COMMENTS

Reviewer #1 (Remarks to the Author):

The authors have addressed my comments/concerns adequately in the revised manuscript.

Reviewer #2 (Remarks to the Author):

The authors have mostly addressed the concerns raised by the reviewers. We appreciate their improved robustness of the results. However, we would recommend, for the data looking at individual cells, that the authors utilize a nested approach (using individual cells as technical replicates within an animal or culture) and use a histogram to show the spread of their data (or other data visualization effort). We appreciate that the field is evolving with this regard, and understand the tension between showing the spread of individual cells (as they are highly heterogenous) and potentially inflating n for statistics; this is why we recommend a nested approach going forward, which takes both into account. There are a few proofreading errors that could be checked for during copy-editing.

Reviewer #3 (Remarks to the Author):

The authors have substantially improved the manuscript and responded to the reviewers' comments. This is a highly interesting study addressing an entire novel aspect in the field of microglia research. I recommend publishing the work as is.

Rebuttal letter v2: Nature Communications manuscript NCOMMS-22-44776A ‘Polarized microtubule remodeling transforms the morphology of reactive microglia cells and drives cytokine release’

*** Authors’ response ***

We thank the 3 reviewers for their final comments and support in publishing the manuscript. We have addressed the remaining minor points in the manuscript.

*** Reviewer #1 ***

The authors have addressed my comments/concerns adequately in the revised manuscript.

*** Reviewer #2 ***

The authors have mostly addressed the concerns raised by the reviewers. We appreciate their improved robustness of the results. However, we would recommend, for the data looking at individual cells, that the authors utilize a nested approach (using individual cells as technical replicates within an animal or culture) and use a histogram to show the spread of their data (or other data visualization effort). We appreciate that the field is evolving with this regard, and understand the tension between showing the spread of individual cells (as they are highly heterogenous) and potentially inflating n for statistics; this is why we recommend a nested approach going forward, which takes both into account. There are a few proofreading errors that could be checked for during copy-editing.

*** Reviewer #3 ***

The authors have substantially improved the manuscript and responded to the reviewers’ comments. This is a highly interesting study addressing an entire novel aspect in the field of microglia research. I recommend publishing the work as is.

Rebuttal letter: Nature Communications manuscript NCOMMS-22-44776-T ‘Polarized microtubule remodeling transforms the morphology of reactive microglia cells and drives cytokine release’

*** Reviewer #1 ***

The manuscript by Adrian and colleagues aims to study the molecular basis of early morphological changes in microglia reactivity. The manuscript is well written and of high significance, including a wider audience outside the neuroimmunology field (see also comments below). The results are novel and support the authors conclusions - with limitations as noted below. In particular, the authors should be commended on their methods section. Early microglia responses, in particular those independent of changes in gene expression have so far been largely unstudied. In this context, it is not clear to me why the authors included the in vivo studies in the Alzheimer's disease models? These are clearly not early reactive changes and the overall microglia response in the brains of these mice is confounded by a complex neurodegenerative and neuro-reactive milieu. The authors also assume that reactive microglia (see also specific comment below in regards to the term 'activated') undergo similar morphological changes in response to stimuli. This is incorrect (e.g. see recent publication by West et al. 2022, J. Neuroinflammation); rather, microglia do show stimulus specific responses which do include clear morphological changes. In light of this, the changes induced by LPS, while an important model, are specific for this stimulus. The manuscript would gain in impact by looking at other stimuli (e.g. do similar changes in morphology and microtubule changes occur in cells stimulated with individual cytokines?); else, at least a caveat needs to be included throughout the manuscript (incl. title).

We would like to thank the reviewer for the thoughtful comment and truly appreciate that the manuscript was overall well received. We have followed the reviewer's suggestion closely to look at different stimuli, including pathological recombinant proteins (Tau and amyloid beta) and cytokines in-vitro and agree that any stimulus cannot be generalized. However, we did find some similarities in the molecular mechanisms that underlie the morphological rearrangements. We have revised the wording about microglial reactivity throughout the manuscript, de-emphasized the data presented in Fig. 1 and integrated the finding more closely with the in-vitro assays. Please see our response to the specific comments below.

Specific comments:

1. I do not see the purpose of the AD mouse model studies. The findings are not novel and the models have very limited correlation with the in vitro model used for all further experiments.

We agree that the AD models did not connect well with the rest of the findings in the original version of the manuscript. We have reduced the data in this section and emphasized that many of these findings are not novel but used to validate the sensitivity of our segmentation analysis. We also took care to de-emphasize the importance of this section in view of the scope of this manuscript and put it better into context. We have now also included in-vitro assays showing morphological responses by microglia to recombinant Tau and amyloid beta proteins. While the morphological response to each stimulus is different, we were excited to see that Cdk1 inhibition attenuated LPS phenotypes as well as those of Tau and amyloid beta fibrils.

2. The purity of microglia is shown on the basis of Iba1 levels, which itself is a marker that is regulated by inflammation. While I agree that the purity is sufficient for the study, it would be helpful to include additional markers, in particular those that are commonly used to characterize mature microglia (e.g. CSF1R, 4D4, P2RY12, TMEM119, SALL1). Such analysis would help establish that despite the culture conditions, these cells are 'true' microglia. This could be extended to identify the identity of the remaining 3% cells (presumably astrocytes and/or fibroblasts?).

We agree with the reviewer that the characterization of primary microglia with Iba1 staining alone is not ideal. We have added data in Supplementary Fig. 2b showing bulk RNA sequencing for our cultures and demonstrating that the gene set score for microglia markers is highest in these cultures. We did not perform single-cell sequencing to identify the ~3% cells of other origins as this population is very small and would be very difficult to determine. Similar primary microglial cultures have been extensively used in the literature and also characterized by our colleagues before (see e.g. [10.1016/j.cell.2020.07.011](https://doi.org/10.1016/j.cell.2020.07.011)) and we agree with the reviewer that such purity should be more than sufficient for our study.

3. Microglia are constantly active; please avoid using the word "activated" or "activation state"; I suggest to use "reactive" or "activity states" instead. Similarly, the term "resting" needs to be avoided.

We thank the reviewers for pointing this out. We have changed the wording as suggested throughout the manuscript, referring to LPS-reactive and non-stimulated microglia where appropriate. We agree that the previous wording is misleading and did not intend to suggest that non-stimulated microglia were inactive.

4. Microglia undergo stimulus-specific morphological changes and the previous assumption of "activated" microglia being amoeboid is just wrong (see also above).

We thank the reviewer for bringing up this important point. Indeed, reactive microglia have diverging morphological phenotypes depending on the stimulus they encountered, the brain region they reside in, and the time they have been exposed to this signal. We have highlighted this now in the introduction and result sections. What we try to convey in Fig. 1 is that, albeit being exposed to different stimuli, microglia exhibit some general changes in their cell morphology that are shared in different activity paradigms. Our morphological analysis does not dig into the branching patterns where most of the morphological diversity can be observed. Instead, we focused on an overall loss of ramification in-situ and a change in cell size in vitro and were looking for underlying cytoskeletal mechanisms facilitating this change.

In response to the comments above, we have now also tied Fig. 1 and Fig. 2 closer together by performing the morphological analysis in vitro with additional stimuli. We show that indeed morphological phenotypes are different for LPS, recombinant amyloid beta and tau proteins or cytokine stimulation. We are also excited to share that Cdk1 inhibition rescued both the increase in cell size elicited by LPS and amyloid beta fibril reactivity as it did for the decreased cell size in response to Tau^{P301S} fibrils. We believe that this data shows that there are common underlying

molecular mechanisms to these activity states that can contribute to specific responses for each stimulus.

5. Figures should show individual values throughout.

We plotted all quantifications with scatters or as boxplots where possible. One exception is proteomic quantifications where the data had been averaged over the replicates in an upstream pipeline. We do, however, provide all means and SE values in the Source Data and supply the raw proteomic output in a publicly available database for these data. Lastly, for the large Luminex panels in the Supplementary Figures, we did not add the scatter for clarity of the figure panels, but provided the individual values of each replicate measurement in the Source Data.

6. It would be good to include more raw data, in particular for the immunoblots (e.g. full gel images)

We fully agree with the reviewer and we have added all raw measurements underlying all graphs as well as uncropped western blots and proteomic and gene ontology data in the Source Data and Supplemental Data files in the revised version. In addition, we deposited the raw proteomic and RNA sequencing data in public repositories, all in accordance with NPG's editorial policies.

*** Reviewer #2 ***

In “Polarized microtubule remodeling transforms activated microglia morphology and drives cytokine release,” Adrian et al., investigate microtubule remodeling in relation to microglia morphology. The authors first highlight their segmentation model for IBA1+ cells in the steady-state adult mouse hippocampus and cortex, and then show they too can detect changes in morphology with their analysis method in various models of neurodegeneration. The authors further highlight their segmentation model in a primary microglia culture system. The authors then show that hyper-stabilizing and depolymerizing microtubules prevents changes in microglia cell culture morphology following LPS. Then, the authors performed a time course of the proteome and phospho-proteome of primary microglia stimulated with LPS, finding that in addition to cytokines, proteins involved in the cytoskeleton and microtubule components were increased. In addition to changes in protein levels, there were changes in protein phosphorylation, including in those not changed in number, including Map4 and Stmn1. The authors then investigate the microglia tubulin, finding that with LPS, there is an increase in radially polarized cells. The authors then demonstrate that STMN1, MAP4 and CDK1 play a role in microglia microtubule polarization in vitro. Finally, the authors exposed ex vivo brain slices to LPS and CDKi, showing that inhibition of CDK1 can attenuate LPS induced reductions in cell size. Overall, there are interesting data presented in the paper, particularly the examination of molecular mediators of microglia microtubule changes. However, aspects of the paper require attention.

We thank the reviewer for these important comments which we have answered in full below. We are grateful to hear that the reviewer shares our enthusiasm for this study. In our revised version we have addressed the low number of replicates in a number of assays, we clarified and documented our statistical tests in more detail throughout the manuscript and addressed the relevance of Figure 1 to the rest of the study. We especially thank the reviewer for this important feedback on the main text related to Figure 1, which we have extensively edited to address these concerns about undervaluing the field, which was absolutely unintentional. We believe that this has strengthened our manuscript and we detail our responses to your comments below.

1. There is concern about the statistics. Statistics should be performed on the whole mouse/independent cultures, not the individual cells. There are numerous cases where the results come from 2 animals or cultures/groups, which is insufficient. A nested statistical approach could be utilized to account for technical replicates from the same biological replicate.

We thank the reviewer for pointing out these important issues. He/she is absolutely correct and we agree that experimental reproductivity, robustness, and transparency are very important. In response, we repeated all assays to increase the number of independent experiments/cultures and report these numbers throughout the manuscript. In particular, we repeated MT+TIP tracking, immunofluorescence stainings, and acute slice cultures. Throughout, this resulted in more robust statistics but did not change our conclusions and interpretations.

For in-situ data, we show all results averaged per animal, as the reviewer requested.

For acute slices, we increased the N of mice from two to five and averaged individual cell measurements into ROIs and then individual slices. Because brain slices rather than mice were the experimental unit and because a N of five is still not statistically powerful enough to detect changes in subpopulations of cells, we plotted the results and performed the ANOVA on brain slices (2-3 per animal). In the Source Data we also provide the dataset per cell and per slice as well as the output of the Tukey HSD test.

For in-vitro assays pursued two approaches based on the type of experiment performed. For morphological screens, we measured hundreds of cells per well and averaged them per well, the experimental unit that was manipulated. For each of these experiments, we identify the number of wells and the number of independent cultures that these cells came from. For experiments, where we measure detailed immunofluorescence stainings or image the dynamics of individual microtubule plus-ends, we report the results per cell rather than averages per cultures. Variation between primary cells in culture is typically larger than batch-to-batch culture differences and we would like to show the spread between individual cells to show robustness of the assay. We provide all raw measurements along with their detailed statistical test outputs from R or Prism in the Source Data and identify cell and culture numbers in the figure legends throughout.

2. The interpretation from culture studies is interesting. a. Morphological changes observed in neurodegeneration in Figure 1 change from highly-ramified to less ramified, could extended ramified processes not travel inward before then radially extending outward? How do the authors know if these are at specific morphologies rather different stages in morphological alterations? Especially given the number of unpolarized cells increases at 1 and 2 hours?

Terrific point - it is very much possible that the cytoplasm and membrane pools 'gained' by retracting the protrusions contribute to the process of growing the cell body. We believe the two processes are linked. In situ, the growth of cell bodies (Supplementary Fig. 1c - Inscribed radius - a measure for cell bodies without protrusions) is typically correlated with smaller overall cell size (a measure taking into account cell bodies and protrusions).

In vitro, our primary cells are already less ramified in their non-stimulated condition. Hence in these assays, we use cell size as a measure to describe the overall flattening and growth of the cells on tissue-culture substrates.

In both cases, we believe that the underlying remodeling of microtubules extending from the centrosome leads to an increase in cell (body) size that requires the cell's protrusions to shrink. We, unfortunately, do not know how this process is maintained or changed over longer time ranges in late pathological conditions in vivo.

2b. Do microglia of the same type of polarization (radially versus unpolarized) in the same group (control versus LPS) have the same directionality of MT growth? In other words, do all microglia in LPS grow outward, regardless of morphology? Or do all unpolarized and other cells grow in outward and inward regardless of condition?

We thank the reviewer for this interesting question. Indeed, cell morphology can be heterogeneous in both control and LPS-stimulated conditions. We did our best to image cells in all conditions without bias and record MT+TIP dynamics from a representative group of cells per treatment condition. Especially at early time points of LPS treatment, the cell morphology

can vary substantially between cells. As quantified in Figure 4e, you can appreciate that a subset of cells still has a fraction inward-growing microtubules and these are indeed seen mostly in less radially-polarized cells. In this assay, we did not express any membrane proteins that could allow us to accurately characterize the cell morphology of each cell we imaged and therefore did not further quantify this correlation.

2c. The CDKi proteomics study (unclear if this was done with whole brains or with cultures), the timeline of cytokines examined is curious. The earlier proteomics data shows pro-inflammatory cytokine increases starting at 0.5h. It is unclear based on the presented data if CDK1 leads to increased pro-inflammatory cytokines in response to LPS; although it could potentially contribute to the sustained increases in pro-inflammatory cytokines. (Similar with the supernatant cytokine levels).

The CDKi proteomics study in Fig. 6 was performed on primary microglia cultures, just as the LPS time course study in Fig. 3. We show the experimental setup in Supplementary Fig. 5e. The CDKi study did not include phospho-proteomic analysis, hence we chose to focus on timepoints at which we would expect protein translation levels to change, 8 and 24h, that we also used in the LPS proteomic study. The 30 min to 4 h time points in the initial study were mainly chosen for phospho-proteomic analysis, as changes in translation and protein abundance usually take several hours to reach maximum effect. Indeed, several cytokines can be picked up earlier as shown in Fig. 3 but protein levels of these cytokines remained elevated at 8 and 24 h. We agree that the experiment does not address whether CDKi inhibits the onset or sustained release of cytokines. However, the Luminex assay in Fig 5k shows the cumulative amount of cytokines released over 24h.

3. Segmentation protocol should be better documented and the steps should be visualized in a figure.

We have mentioned all steps of both segmentation protocols in the Materials and Methods section and also now added a panel summarizing the steps visually in Supplementary Figure 1a.

4. Overall, there is imprecise language throughout the manuscript:
• Please avoid the phrasing, “microglia activation,” and name specific processes or responses as per field guidelines recommended in Paolicelli et al., 2022, Neuron. See also “resting” microglia.

We thank the reviewer for addressing these inaccuracies. We have changed the language about microglial reactivity and stimulations throughout the manuscript and believe that this better and more accurately reflects our experiments now.

• Line 104 – ex vivo or in situ rather than in vivo. All subsequent tissue work should be referred to as such

We have now clearly distinguished between in vitro in-situ, acute slices and in-vivo processes throughout the manuscript.

- The tone of this manuscript, particularly at the beginning seems to ignore much of the microglial field. This reviewer understands that manuscripts need to “sell” novelty but this should not be at the expense of building on the current field. Segmentation approaches exist. Microglia morphological changes have been demonstrated in the models utilized by the authors. There is interesting data in the manuscript that is underserved by the framing of the first parts of the results. For example, consider, validation of novel morphological workflow in microglia across neurodegenerative disease models rather than, “Morphological microglia changes are a quantifiable hallmark in neurodegenerative models,” which implies a novelty to the results when this type of work has been shown before. Lines 100 and 140 also have a similar statement.

We thank the reviewer for this candid feedback. It was not our intention to oversell our morphological analysis. Indeed, there are many algorithms identifying microglial morphologies at different levels of detail and technical sophistication, several of which we mention in this section. We have rewritten this section extensively to remove any suggestion of inventing a first-of-kind analysis. The strength of our approach lies in its relative simplicity with enough sensitivity to detect changes in cell morphology of common Iba1-DAB stainings. We also took care to de-emphasize the importance of this section in view of the scope of this manuscript and connect it more directly to our in-vitro data by adding additional stimulations to our panel in Fig. 2a.

Minor clarifications

- Line 333, attenuated rather than robust rescue (were comparisons made between the control and CDKi?)

We thank the reviewer for this suggestion and have altered the text accordingly.

- Standard housing conditions are a 12/12 light/dark cycle. Given that there is evidence of circadian changes in microglia, do you think this affected the results? Furthermore, please specify when animals were utilized in their light cycle for experiments.

Our mouse facilities run on a 14/10 hour light/dark cycle as described in the methods section. Since we do not shift the circadian rhythm of the animals, we do not believe there to be an effect of that. All mice were handled at the same time points for injections (mornings, ~1 h post light on). Mice sacrificed for primary cultures were always collected at mid day, 2 days after arrival in their holding rooms.

- Sex of the mice could be better clarified throughout the methods and manuscript.

Sex of mice is now indicated clearly in the methods section for each strain. We did not expect to see differences between sexes based on literature (10.1002/glia.24427) and did not in studies containing both males and females. The sex of each mouse is indicated in the Source Data tables for panels in Fig. 1.

- Jackson Laboratories, not Jaxon, I believe.

We thank the reviewer for this correction and have altered the text accordingly.

- If the methods could follow the order of the results, that would be helpful to the reader.

We thank the reviewer for this suggestion and have altered the text accordingly.

- Please include all secondary antibodies used and provide details for them and the DAB IBA1 (dilution, etc) in the corresponding methods sections.

We have added product details and catalog numbers for secondary antibodies in the revised manuscript. We also added a complete protocol for the Iba1 histochemistry performed by our CRO.

- Imaging details/parameters missing from DAB methods sections

We now added a complete protocol for the Iba1 histochemistry performed by our CRO in the methods section.

- The authors use broad regions to perform their analyses. Can they confirm that the data compared is from the same “region” i.e. always from the same specific subpart(s) of the hippocampus or cortex? There are noted dorsal/ventral, layer and sub-region differences of microglia that have been extensively reported. For example, Figure 1h-l, representative images are taken from smaller areas of the hippocampus or cortex, but was the entire hippocampus imaged, or just a subset of ROIs? See also Figure 7.

Yes, all regions in Fig. 1 were quantified as shown in Fig. 1a. All ROIs were manually curated to encompass similar cortical and hippocampal regions at similar sectioning depth. By eye, the cortical regions appeared very homogenous in control mice while the layering of the hippocampus could be appreciated. We would also like to stress that all quantifications in Fig. 1 were carried out in the entire regions marked by red boxes in Fig. 1a and not on the zoomed images. We measured thousands of cells per ROI that were subsequently averaged per slice and animal. We have highlighted this now in the updated figure legend. In Fig. 7, we show the whole field of view that we analyzed in panel b. For all of these quantifications, we now also supply all raw measurements in the Source Data.

- For the primary cultures, there are known developmental changes to microglia, so do the authors think that the comparison to the adult condition are limited?

We thank the reviewer for bringing up this important topic. Absolutely, we agree that cultured primary microglia from mouse pups are indeed different from adult cells in situ. We addressed these differences in the text. Importantly, as mentioned in the discussion, our observations in the primary cells fit with data known from other in vitro and in vivo studies. We believe that the power of molecular manipulations in primary cells is important to study the underlying mechanisms that cannot be easily addressed in situ even if they are limited by mimicking only certain aspects of microglial morphology and behavior. We hope that future research will bridge

this gap by developing more refined culture methods that better reflect the cells' environment in situ as we mention in the Discussion.

• Do the authors have images of IBA1 in addition to alpha-tubulin for the cell culture studies, to map microtubule phenotypes to microglia morphologies (Figures 4 and 5).

We outlined the cell morphology based on unspecific binding of the antibody for illustrative purposes in the figure but do not have a specific staining that we can use for quantitation. Unfortunately, we cannot generate a direct comparison of cell shape and microtubule staining with our datasets.

• Consider adding scale bar information to images

We provide scale bar information for all microscopy images in the figure legends and have standardized their sizes as much as possible throughout this manuscript.

• Does blocking phosphorylation of STMN1 alter inward dynamics? Similar with MAP4 siRNA and CDKi studies?

siMap4 treatment, Stmn1^{2xSA} overexpression and CDKi treatment does not alter the microtubule growth direction significantly. We have added new data to all MT+TIP tracking experiments and updated the panels in main and supplementary figures. We believe that this effect is mainly due to centrosomal activation that we would only expect to be altered by CDKi. We have not investigated if Cdk-1 inhibition has an effect on Golgi-localized microtubule nucleation. This is the main driver for inward growing microtubules in non-stimulated microglia.

• It would have been interesting to see what the various inhibitors of STMN1, MAP4 and CDK1 did to non “activated” microglia. Not including this data reduces the picture of these pathways in regulating inward growing.

We agree with the reviewer that this is an interesting question. We had addressed this partially in the initial version in key experiments, like the cell morphology assays (Supplementary Fig 5a,b) and in the acute slices experiment (Fig7 a,b). We now also show non-LPS stimulated data in other assays, e.g. in MT+TIP assays. In addition we now express the immunofluorescence levels in Fig 6 relative to non-stimulated control cells (Fig6 d,g,h,j).

• What is normalized cell size used? Especially as it hides the results of the ability of CDKi to reduce LPS induced increases in cell size? More problematic is that using the non-normalized ramification index shows potentially no increases with CDKi compared to controls and an actual reduction in ramification with LPS?

We used normalization to compare cell morphologies between experiments. As batches of primary cells differ, we set the average of the control condition to 1. This still allows to see the spread of the datapoints within each group and to examine the effect of treatments on the cell size and more conveniently shows the effect size of the various experimental conditions. This metric alleviates some batch effects that affected overall cell size of a given culture.

- The authors should consider adding ramification index in addition to cell size, to better compare their ex vivo slice data to the data in Figure 1 and better relate to morphological changes discussed in the paper.

We agree with the reviewer that a combination of these measures gives the best description of the morphology phenotype. We now show ramification index and cell size for all experiments either in primary or supplemental figures. As we mention in the main text, we do find that in vitro, cell size is the most sensitive measure as primary microglia in monocultures are often not very ramified, but elongated and small, in non-stimulated conditions. We mention this limitation in the results and discussion sections and hope that the field will establish better culture conditions for microglial cells to mimic cell ramification more accurately in vitro.

- Keeping scales the same throughout could be helpful for the reader (e.g. Figure 7 c and d, mm³ versus um³).

We thank the reviewer for this suggestion and have changed the figure accordingly.

*** Reviewer #3 ***

In this manuscript by Adrian et al, the authors characterise novel functions of the microtubule cytoskeleton system during the process of microglia activation. They establish a segmentation workflow of Iba-1+ microglia in mouse brain sections in order to quantify morphological changes in different murine neurodegenerative models. From this method development, the authors establish an in vitro model of microglia cell activation involving LPS treatment, which is scrutinised in detail for activation-associated expression of pro-inflammatory markers, for cytokine secretion, for proliferation, as well as for changes in microglia cell size and ramification. Here, first analyses of microtubule function during LPS induced activation on changes in morphology and cytokine secretion is provided. By the use of proteomic profiling of these cultured microglia in the absence and presence of LPS, the authors then identify ‘microtubule remodelling pathways’ that correlate with distinct changes in microglia cell morphology and organisation of their microtubules following LPS-induced microglia activation in vitro. In vitro and in vivo, the authors suggest CDK-1 as critical upstream regulator of microtubule remodelling and morphological changes. Overall, the work provides substantial amount of novel data, focussing on a rather understudied but highly relevant topic, the cellular mechanism of microglia activation. While the data provide significant fundamental information, they are highly relevant in the context of inflammation associated with different brain diseases and conditions. However, the authors need to revisit some of their experiments and tone down several of their statements, especially with regard to in vitro data.

We would like to thank the reviewer for these very thoughtful comments on our manuscript and appreciate that our findings were well received. We have taken great care to address your concerns in our revised version. We validated our segmentation approach with Tmem119 staining in slices adjacent to Iba1 staining, revised wording around microglial activity states and increased statistical transparency throughout the manuscript. Please find our detailed responses to your comment below.

- Hypertrophy is widely used as established hallmark of microglia reactivity in vivo. With this in mind, it is not clear if the method presented in Figure 1 clearly advances from previous work using hypertrophy as activity readout. It could have been useful to compare both methods to report on microglia reactivity. That said, the presented data and methods are compelling and very nicely executed using different neurodegeneration mouse models. A drawback of this figure, however, is the selective use of Iba1 for the identification of microglia. Iba1 is definitely upregulated in reactive microglia, however, many quiescent microglia may well be Iba1- (i.e. Hendrickx et al., 2017; <https://pubmed.ncbi.nlm.nih.gov/28601280/>). Therefore, the authors should include morphometry using additional markers, i.e. identification of TMEM119+/Iba1+/- microglia +/- neuroinflammatory conditions.

Indeed Figure 1 confirms that microglia react in known models of neurodegeneration. Here we demonstrate that our analysis pipeline picks up these changes in cell morphology and allows us to quantify the phenotypes reliably. There are several useful algorithms described in the literature that analyze microglial morphology with various approaches and levels of detail. Our approach is not meant to be superior to them, rather it is a convenient and purpose-built way to

analyze widely available Iba1-DAB-stained tissue. We agree that benchmarking sensitivity of various approaches under standardized conditions for several microglial markers would be an interesting and worthwhile endeavor. However, such comparison would require a ground truth annotation that is not feasible to perform on thousands of cells per ROI and divert strongly from the scope of the current manuscript. We have described our approach in detail and would gladly contribute to such an effort in a follow-up study.

We developed this algorithm specifically for Iba1-DAB staining as this protocol results in darkly stained cell bodies and lighter protrusions. This helps us to identify individual cells in a first step and then lets us use watershedding to find cell borders between their connected protrusions. We would also like to point out that Iba1 cell density, under this staining protocol, does not strongly change in the neurodegenerative mouse models (Supplementary Fig 1a). We agree that we might miss certain microglial populations by only looking at Iba1 staining. However, even in this population a clear morphological response can be measured between non-stimulated and stimulated microglia. In vitro, we used a cell mask staining that stains all cells irrespective of Iba1 expression.

Nevertheless, we agree that Iba1 is regulated during microglial reactivity. We have now applied our algorithm to wildtype mice stained for Iba1 and Tmem119 in adjacent slices and found no statistical differences in cell number and ramification index (Supplementary Fig 1b). Detection of Tmem119 with this particular algorithm was hard to parameterize with the given thresholds, as cell bodies can be missed and we had to substantially change the threshold parameters. We mention this limitation in the main text.

- Microglia cultured in FBS rich medium already show some basal activity, but their reactivity can be further enhanced (as shown in the manuscript). Therefore, the graph in S2 b does not show 'zero microglia reactivity'; similarly, the authors need to tone down the text accordingly.

We unreservedly agree with the reviewer that no microglial population has 'zero activity' - especially in culture - and we did not mean to suggest this with this panel. We have rephrased the passage in the text as we aim to demonstrate that primary microglia can be further polarized into more inflammatory or more anti-inflammatory pathways from their basal activity in vitro. This experiment is meant to highlight that these cultured microglial cells have the capacity to react to strong stimuli with specific transcriptional responses, not that unstimulated cells are inactive. In addition, we have changed the language throughout the manuscript to better reflect the specific stimulations used in the experimental assays, rather than referring to 'activated' microglia.

- Figure 2F, G, H: Did the authors analyze cells present in comparable densities? High densities severely affect morphology in microglia. Analyses of low density cultures +/- LPS, should be performed, ideally during live cell imaging before and during LPS incubation.

We agree with the reviewer that cell density has a strong effect on microglial reactivity. Before starting the assays described here, we standardized seeding densities and kept their numbers comparatively low and constant. We listed the seeding densities in the methods section for

reference. It should also be noted that in our culture conditions, we did not see cell proliferation, keeping the cell numbers constant throughout the course of these experiments (1-3 days).

- I am somewhat puzzled by the description of ‘radially polarized’ cells. Seems to be an oxymoron. The authors need to comment.

We thank the reviewer for pointing out this inaccuracy. We distinguish two microtubule phenotypes in our cultures. In non-stimulated microglia, we tend to see more cells with dense, tangled and disorganized microtubule networks. In LPS-stimulated microglia we mostly see radial microtubules arrays emanating from the MTOC towards the cell periphery. We have now clarified the language in the text to clarify that we are referring to radial microtubules rather cell polarization.

- CDK1 inhibitor RO3306: working concentration of 10 μ M is high. RO3306 blocks CDK1 and other targets already in the nM range, therefore, off target effects are likely

We thank the reviewer for addressing this important topic. The reviewer is correct that lower concentrations of RO-3306 are sufficient in in vitro experiments with purified kinases. However, for cell-based assays, the standard in the literature is 10 μ M. Many studies on mitotic cell signaling use 10 μ M RO-3306 to inhibit Cdk1 in cell culture. In addition, we titrated RO-3306 and roscovitine in our assay in Supplementary Fig5a, b. We see a dose dependent attenuation of the LPS phenotype starting at 3 μ M RO-3306. We also see a dose-dependent effect of RO-3306 on cytokine secretion in Fig. 6k and Supplementary Fig. 5k that partially inhibits release at 5 μ M. We do agree that follow-up studies may be needed to test the dependence on microglia on Cdk-1 signaling genetically. Since Cdk-1 is an essential gene, we would suggest studying this in a conditional Cdk-1 KO restricted to microglial or macrophage populations.

- Most figure legends do not contain sufficient/any information on experimental and statistical parameters

We agree with the reviewer that transparency about statistical testing is very important. We added n, and variance identifiers to all figure legends consistently. We now also state the type of statistical comparison in the figure legends. For simplicity, we show p-value groups with stars in the figure panels but include exact p-values and statistical output from R and Graphpad into the Source Data file for each individual panel. We also expanded the Methods section on statistical analysis.

- The title suggests that the paper focuses on the process of cytokine release, which is investigated mainly ‘indirectly’ through measurement of cytokine secretion into the supernatant. Should be changed to reflect the main findings of the study.

We would like to point the reviewer to our data on TNF α transport and distribution in microglia cells (Fig. 6j, Supplementary Fig. 6j). We show that LPS stimulation strongly increased TNF α staining, but this was attenuated in reactive microglia also treated with CDKi. Moreover, TNF α -positive vesicles were clustered more closely in the perinuclear region and penetrated less into the cell periphery. In addition to the Luminex cytokine secretion assay results, these data stress

the importance of the microtubule-dependent re-organization in reactive microglia for cytokine transport, such as TNF α to the plasma membrane for exocytosis.